# Ferromagnetic quasi-atomic electrons in two-dimensional electride

Seung Yong Lee [1,2,10], Jae-Yeol Hwang [1,3,10], Jongho Park [1,2], Chandani N. Nandadasa [4], Younghak Kim[5], Joonho Bang[1], Kimoon Lee[6], Kyu Hyoung Lee [7], Yunwei Zhang [8], Yanming Ma[8], Hideo Hosono[9], Young Hee Lee [2], Seong-Gon Kim [4] & Sung Wng Kim[1✉]

An electride, a generalized form of cavity-trapped interstitial anionic electrons (IAEs) in a positively charged lattice framework, shows exotic properties according to the size and geometry of the cavities. Here, we report that the IAEs in layer structured $[Gd_2C]^{2+} \cdot 2e^-$ electride behave as ferromagnetic elements in two-dimensional interlayer space and possess their own magnetic moments of ~0.52 $\mu_B$ per quasi-atomic IAE, which facilitate the exchange interactions between interlayer gadolinium atoms across IAEs, inducing the ferromagnetism in $[Gd_2C]^{2+} \cdot 2e^-$ electride. The substitution of paramagnetic chlorine atoms for IAEs proves the magnetic nature of quasi-atomic IAEs through a transition from ferromagnetic $[Gd_2C]^{2+} \cdot 2e^-$ to antiferromagnetic $Gd_2CCl$ caused by attenuating interatomic exchange interactions, consistent with theoretical calculations. These results confirm that quasi-atomic IAEs act as ferromagnetic elements and trigger ferromagnetic spin alignments within the antiferromagnetic $[Gd_2C]^{2+}$ lattice framework. These results present a broad opportunity to tailor intriguing ferromagnetism originating from quasi-atomic interstitial electrons in low-dimensional materials.

[1] Department of Energy Science, Sungkyunkwan University, Suwon 16419, Republic of Korea. [2] Center for Integrated Nanostructure Physics, Institute for Basic Science, Suwon 16419, Republic of Korea. [3] Department of Physics, Pukyong National University, Busan 48513, Republic of Korea. [4] Department of Physics & Astronomy and Center for Computational Sciences, Mississippi State University, Mississippi State, Mississippi 39762, USA. [5] Pohang Accelerator Laboratory, Pohang University of Science and Technology, Pohang 37673, Republic of Korea. [6] Department of Physics, Kunsan National University, Gunsan 54150, Republic of Korea. [7] Department of Materials Science and Engineering, Yonsei University, Seoul 03722, Republic of Korea. [8] State Key Laboratory of Superhard Materials & Innovation Center of Computational Physics Methods and Software, College of Physics, Jilin University, Changchun 130012, China. [9] Materials Research Center for Element Strategy, Tokyo Institute of Technology, Yokohama 226-8503, Japan. [10] These authors contributed equally: S.Y. Lee, J. Y. Hwang. ✉email: kimsungwng@skku.edu

nterstitial electrons occupying structural cavities have been studied extensively in many condensed matter systems, such as trapped electrons in vacancy defects of crystalline solids and solvated electrons in molecular clusters of polar liquids[1,2]. The electrons often compose ionic crystals as structural ingredients together with counter-cationic molecules or lattices and form an electride, in which interstitial electrons behave as anions in structural cavities[3–5]. In terms of electride functionality, three factors associated with interstitial electrons should be considered critical: concentration, cavity size, and geometry. The interstitial electrons trapped in vacant sites of ionic crystals, such as negatively charged color centers, hardly affect the physical properties, except for optical luminescence[6]. Although the concentration of a color center exceeds the order of $10^{20}$ cm$^{-3}$, a center is merely regarded as a trapping state due to strong localization in small spaces comparable to an atomic size of ~1 Å, inevitably resulting in a deep energetic state that is irrelevant to the active functionality of electrical conduction and magnetic ordering. However, as observed for a metal–insulator transition in alkali metal–ammonia solutions[2], interstitial solvated electrons occupying a large space of ~4 Å surrounded by ammonia molecules can percolate according to their concentration, exhibiting a delocalized state in disordered irregular arrays.

Furthermore, the anionic electrons occupying ordered interstitial crystallographic sites have shown exceptional functionalities according to their degree of localization as well as their geometry[7,8]. When interstitial anionic electrons (IAEs) form an electride crystal occupying structural cavities with a typical size of 4–5 Å, the correlation between IAEs primarily imparts the functionality of electrides. Indeed, the inorganic electride, $[Ca_{24}Al_{28}O_{64}]^{4+}\cdot4e^{-}$ exhibits various electronic phases, from insulating to superconducting, in which the IAEs are trapped in a zero-dimensional cage structure and gradually delocalize with an increase in their concentration[9,10]. It was recently reported that the layer structured $[Ca_2N]^{+}\cdot e^{-}$ electride showed an extremely high mobility as well as distinct anisotropic properties in electrical transport and work function, which were ascribed to the fully delocalized high-density IAEs ($1.39 \times 10^{22}$ cm$^{-3}$) confined in two-dimensional (2D) interlayer space. The layer structured $[Ca_2N]^{+}\cdot e^{-}$ electride was referred to as "2D electride"[8], triggering the development of various 2D electrides[11–14].

Recent theoretical studies have predicted that many elements can transform into electrides under high pressure[15–19]. Some of the predicted alkali metal electrides have been experimentally observed[20]. The strongly localized IAEs in the cavities of elemental electrides are generalized and elaborated as interstitial quasi-atoms (ISQs), which fill the quantized orbitals of the interstitial sites enclosed by the surrounding atom cores[5,15]. However, experimental evidence for quasi-atomic IAEs is yet to be discovered in practical electrides. Beyond the context of structural predictions for elemental electrides, diverse electronic phase transitions, from metallic to insulating or superconducting states, have been successfully demonstrated under high pressure[21]. In particular, dense potassium adopts a stable open structure with strongly localized IAEs in a specific crystallographic site, exhibiting a Stoner-type instability towards s-band ferromagnetism based on the IAEs[16]. The ferromagnetic (FM) instability is also expected in the 2D layer structured inorganic $[Y_2C]^{2+}\cdot2e^{-}$ electride[22]. Furthermore, it has been demonstrated that the localized IAEs of several organic electrides interact with each other through cavities in a one-dimensional (1D) channel, rendering a weak magnetic ordering such as antiferromagnetism[4]. It is further predicted that the antiferromagnetic (AFM)–FM transition in the simplest organic Cs$^{+}$(15-crown-5)$_2\cdot e^{-}$ electride can be realized under an easily accessible experimental condition (0.5–1 GPa) which enables a strong spin coupling of localized

IAEs[23]. However, to date, a FM electride has not been discovered in experiments.

Considering the antiferromagnetism and possible ferromagnetism of organic electrides originating from IAEs in 1D channels together with the anisotropy of electronic states[4,23,24], it is possible to realize a FM electride based on strongly localized IAEs occupying a regular array in interlayer space of layer structured 2D electride. If a 2D electride is composed of elements with a high valence state and anisotropic d- or f-orbitals that result in a strong electrostatic attraction with anionic electrons, leading to the strong localization of IAEs, we expect that such IAEs in 2D electrides can behave as quasi-atomic FM particles facilitating magnetic ordering under anisotropic spin fluctuation. In this study, we report that the strongly localized IAEs in layer structured $[Gd_2C]^{2+}\cdot2e^{-}$ possess their own magnetic moments and behave as FM elements. Furthermore, the quasi-atomic IAEs promote the exchange interactions between Gd atoms across IAEs, which emerges FM ordering within the AFM $[Gd_2C]^{2+}\cdot2e^{-}$ lattice framework.

## Results

**Synthesis and structural characterization.** Quasi-atomic IAEs with their own magnetic moments are found in the interlayer space of the 2D $[Gd_2C]^{2+}\cdot2e^{-}$ electride. The crystal structure of digadolinium carbide, $[Gd_2C]^{2+}\cdot2e^{-}$ (Fig. 1a), which is an anti-CdCl$_2$-type layered structure belonging to the $R\bar{3}m$ space group, was determined from single-crystal analysis (Fig. 1b–d) and Rietveld refinement of the corresponding powder X-ray diffraction pattern (Fig. 1e; Supplementary Table 1). The $[Gd_2C]$ layer unit is formed by edge-sharing Gd$_6$C octahedra and is separated by ~3.38 Å along the c-axis. The X-ray absorption spectra (XAS) of single-crystal $[Gd_2C]^{2+}\cdot2e^{-}$ (Fig. 1f, g) show that the valence states of Gd and C are +3 and −4, respectively. This finding indicates that the 2D layered lattice framework is a stacked structure composed of positively charged $[(Gd^{3+})_2C^{4-}]^{2+}$ cationic slabs. Considering that the Gd–C distance is 2.52 Å, each $[(Gd^{3+})_2C^{4-}]^{2+}$ slab is composed of mixed ionic and covalent bonds between Gd$^{3+}$ and C$^{4-}$, limiting the interstitial space for the two excess electrons (2e$^{-}$) to serve as counter-anionic electrons in the cationic slabs (Supplementary Fig. 1c). Thus, the interlayer space between cationic slabs functions as an interstitial anionic site for two excess electrons, resulting in the chemical formula of $[Gd_2C]^{2+}\cdot2e^{-}$ as a layer structured 2D electride, analogous to the previously reported 2D $[Ca_2N]^{+}\cdot e^{-}$ and $[Y_2C]^{2+}\cdot2e^{-}$ electrides[8,11,13].

While a common feature among the three 2D electrides is that the outermost orbital character of neighboring atoms interacting with IAEs is the metal cations, a critical difference from the two previously reported 2D electrides should be noted for $[Gd_2C]^{2+}\cdot2e^{-}$: the interlayer space (~3.38 Å) is shorter than that of $[Ca_2N]^{+}\cdot e^{-}$ (~3.86 Å)[8] and similar to that of $[Y_2C]^{2+}\cdot2e^{-}$ (~3.29 Å)[13]; consequently, the IAEs in $[Gd_2C]^{2+}\cdot2e^{-}$ are strongly localized in the interlayer space in contrast to the fully delocalized IAEs in $[Ca_2N]^{+}\cdot e^{-}$ and similar to those in $[Y_2C]^{2+}\cdot2e^{-}$ (Supplementary Figs. 1d–f). In contrast to $[Ca_2N]^{+}\cdot e^{-}$, with fully delocalized 2D IAEs in the interlayer space, the IAEs of $[Gd_2C]^{2+}\cdot2e^{-}$ are found to occupy the crystallographic Wyckoff position of (0, 0, 0.5) between cationic slabs (Fig. 1a) due to the strong localization originating from the interaction with the anisotropic d-orbitals of Gd, which is discussed in a later section.

**FM properties.** The strong localization of s-like IAEs interacting with Gd$^{3+}$ ions that arrange seven 4f electrons in a half-filled shell corresponding to an all-parallel spin with $S = 7/2$ provides highly anisotropic magneto-transport properties. Figure 2a shows the anisotropic transport behavior of bulk single-crystal $[Gd_2C]^{2+}\cdot2e^{-}$

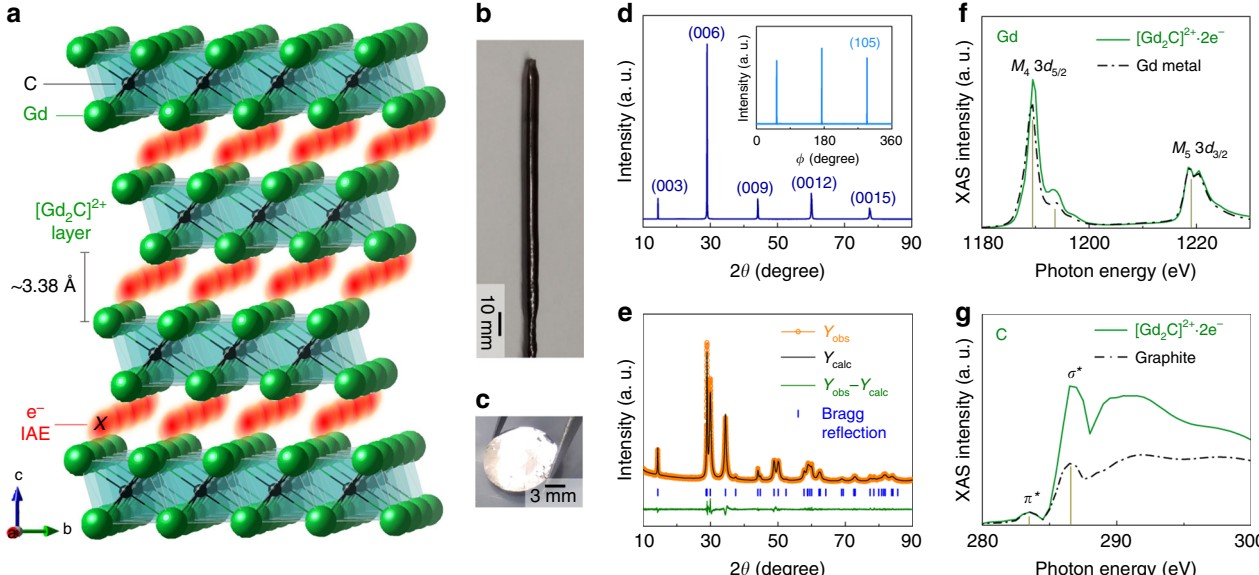

**Fig. 1 2D [Gd₂C]²⁺·2e⁻ electride with strongly localized IAEs. a** Schematic illustration of the crystal structure of [Gd₂C]²⁺·2e⁻ electride with IAEs in the interlayer space between [Gd₂C]²⁺ layers. **b**, **c** Photographs of single-crystal [Gd₂C]²⁺·2e⁻ electride (**b**) and its cleaved surface (**c**). **d** 2θ scan of the cleaved surface. The inset is the ϕ scan for the cleaved surface, indicating the well-constructed rhombohedral unit cell with threefold symmetry. **e** Rietveld refinement of XRD patterns for pulverized powders of a single-crystal. The detail results are given in the Supplementary Table. 1. **f**, **g** XAS spectra of Gd (**f**) and C (**g**) for single-crystal [Gd₂C]²⁺·2e⁻ electride. Dashed curves are the spectra of Gd metal and graphite.

electride, in which the metallic conduction along the out-of-plane is suppressed by a factor of ~60 compared with that along the in-plane direction over the entire temperature range of 2–400 K. According to the power-law fitting ($\rho(T) = \rho_0 + AT^n$) for the temperature ($T$) dependence of electrical resistivity ($\rho$), $n$ values of 0.8 and 1.5 over the range 40 K < $T$ < 350 K were obtained in the out-of-plane and in-plane directions, respectively, indicating that both behaviors are ascribed to the scattering of itinerant electrons with magnetic spins[25] (Supplementary Fig. 2a). A clear transition is observed at approximately 350 K for both directions, as indicated by arrows (Supplementary Fig. 3a). Hall effect measurements indicate an anomalous magnetic field ($H$) dependence of the Hall effect below 350 K, which is characteristic of FM materials (Supplementary Fig. 2b). The carrier concentration ($N_e$) estimated from the linear $H$ dependence of the Hall effect assuming the free electron model is ~$2.9 \times 10^{22}$ cm⁻³ at 300 K, which is similar to the theoretical $N_e$ of $2.84 \times 10^{22}$ cm⁻³ based on the chemical formula of [Gd₂C]²⁺·2e⁻.

Figure 2b, c shows the magnetic properties of [Gd₂C]²⁺·2e⁻ electride. The temperature dependence of magnetization ($M$) under different $H$ directions confirms that the observed transition in resistivity at 350 K is due to the FM transition. While the FM critical temperature (Curie temperature, $T_C$) is identical in both directions, the $M$ of the in-plane direction is saturated at lower $H$ than that of the out-of-plane direction, indicating that the easy and hard axes are correlated with the in-plane and out-of-plane directions, respectively. We note that the magnetic moment of [Gd₂C]²⁺·2e⁻ electride (15.6 μB) is greater than the corresponding value of the seven fully aligned half-filled 4f electrons with $S = 7/2$ in the [Gd₂C]²⁺ lattice framework ($2 \times 7$ μB = 14.0 μB) or even that of Gd metal ($2 \times 7.6$ μB = 15.2 μB)[26]. In addition, the $T_C$ of [Gd₂C]²⁺·2e⁻ electride (350 K) is higher than that of Gd metal (293 K)[27], indicating that an extra contribution over FM Gd metal is required to account for the FM ordering in [Gd₂C]²⁺·2e⁻ electride.

The $H$ dependence of magnetoresistance (MR) for FM [Gd₂C]²⁺·2e⁻ electride (Fig. 2d) is significantly different from that of typical FM materials[28,29] and reported 2D electrides. The variation of the

MR ratio showed anisotropy with respect to the direction of the applied $H$, decreasing continuously at 40 K with a kink at each saturation field ($H_s$), as indicated by arrows. However, the MR ratio at 2 K decreased up to $H_s$ and then slightly increased due to the magnetic Lorentz force over $H_s$[28,29], exhibiting the characteristic MR feature of FM/nonmagnetic (NM) multilayered systems[30,31]. X-ray magnetic circular dichroism (XMCD) measurements (Fig. 2e) provide a plausible explanation for the intriguing MR behavior in FM [Gd₂C]²⁺·2e⁻ electride. A clear XMCD signal was observed at the Gd $M_5$ edge, with a deep negative minimum and a low-energy positive peak at the Gd $M_4$ edge, while the signal from the C $K$ edge was negligible, as shown in the inset of Fig. 2e. This result indicates that the C atomic layer magnetically operates as an NM layer sandwiched between the virtual FM layers, which consist of two Gd atomic layers interacting via IAEs in the interlayer space. Thus, it is hypothesized that the virtual FM layer is formed by the exchange interaction in the Gd–IAE–Gd linkage, as schematically illustrated in Fig. 2f. Furthermore, the MR feature implies that the exchange coupling between virtual FM layers across the NM layer exists, leading to a consideration of three-dimensional (3D) ferromagnetism of [Gd₂C]²⁺·2e⁻ electride with anisotropic magnetic properties.

**Quasi-atomic nature of IAEs.** To elucidate the role of IAEs on the exchange interaction in the Gd–IAE–Gd linkage giving rise to the intriguing ferromagnetism of [Gd₂C]²⁺·2e⁻ electride, we investigated the electronic and magnetic spin structures by ab initio calculations based on spin-polarized density functional theory (DFT). The electronic band structure and density of states (DOS) are shown in Fig. 3a, b, respectively. The DOS for Gd-$f$, Gd-$d$, and IAE-$s$ orbitals (Fig. 3c–e) and projected band structure (Fig. 3f, g) reveal that the densely populated and nearly dispersionless bands lying more than 4 eV below the Fermi energy ($E_F$) are derived from the unpaired electrons of Gd-$f$ orbitals, and the conduction electrons occupying the bands within 1 eV of the $E_F$ mainly originate from Gd-$d$ and IAE-$s$ orbitals (Fig. 3d, e). The electron localization function[32] (ELF) plots for the majority and

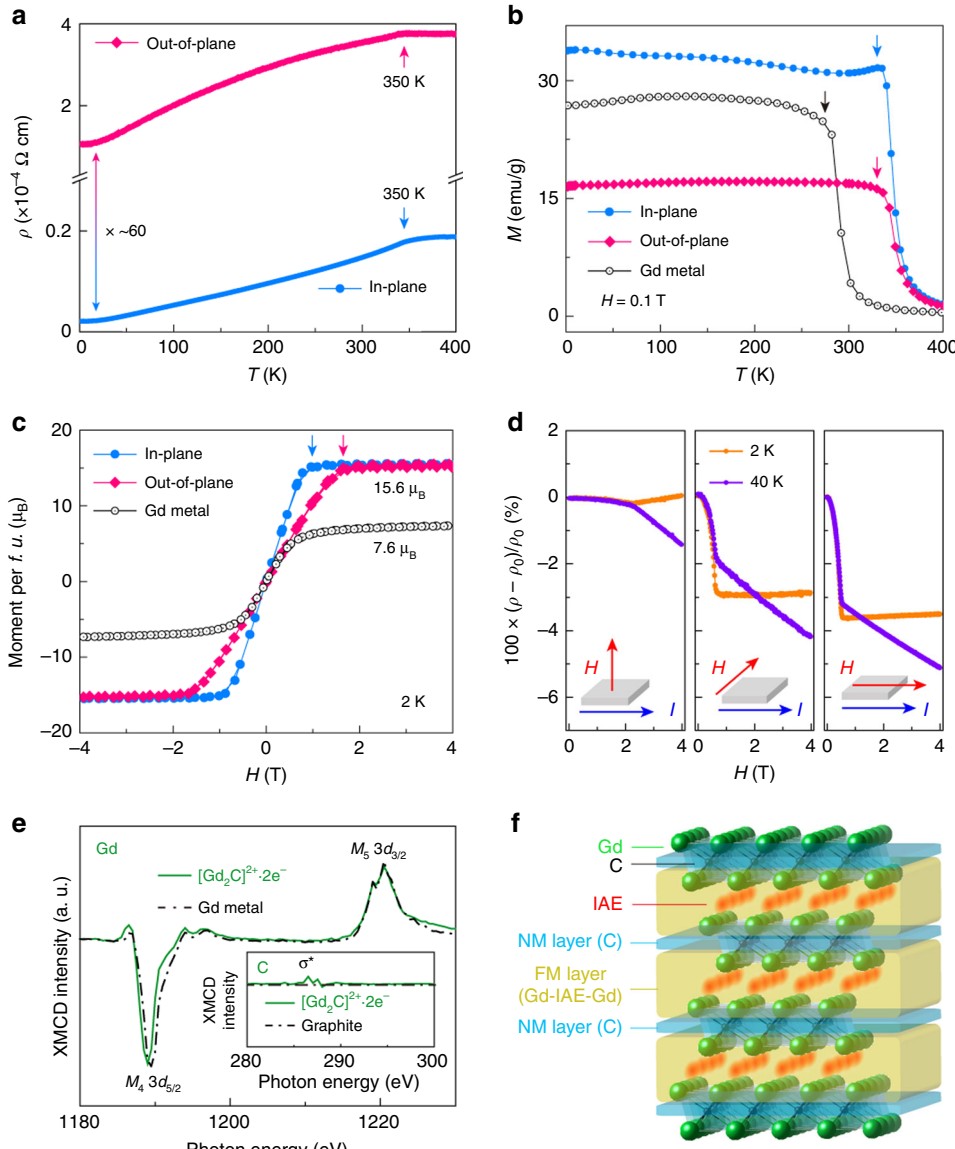

**Fig. 2 FM 2D [Gd$_2$C]$^{2+}$·2e$^-$ electride. a, b** Temperature ($T$) dependences of electrical resistivity ($\rho$) (**a**) and magnetization ($M$) (**b**) for single-crystal [Gd$_2$C]$^{2+}$·2e$^-$ electride and Gd metal under a magnetic field of 0.1 T. **c** Magnetic field ($H$) dependence of $M$ for single-crystal [Gd$_2$C]$^{2+}$·2e$^-$ electride and Gd metal at 2 K. Each arrow indicates the saturation field of $M$. **d** $H$ dependence of magnetoresistance under different directions of $H$. **e** XMCD spectra for Gd and C (inset) of single-crystal [Gd$_2$C]$^{2+}$·2e$^-$ electride compared with Gd metal and graphite (dotted curves). **f** Schematic illustration of the proposed FM/NM layered magnetic structure.

minority spins indicate that IAEs are strongly localized at the position marked as "$X$" in the interlayer space (Fig. 3h, i).

The comparison of the orbital energies between [Gd$_2$C]$^{2+}$·2e$^-$ electride and the [Gd$_2$C]$^{2+}$ lattice framework shows that the energy of the IAE-$s$ orbital of [Gd$_2$C]$^{2+}$·2e$^-$ electride is ~0.4 eV lower than that of the Gd-$d$ orbital of the [Gd$_2$C]$^{2+}$ lattice framework (Supplementary Fig. 4). Thus, the localization of $d$-electrons of Gd into $s$-electrons of IAEs stabilizes the system as an embodiment of an ISQ[1]. The main part of the projected band structure for IAEs is mostly flat, indicating that the IAEs are strongly localized, as confirmed from the plots of ELF. By integrating the charge density within Bader's basin containing IAEs[33], we obtained an atomic charge of 1.8e$^-$ for each localized position of IAEs, which is close to the nominal value of 2e$^-$ (Supplementary Table 2). These results validate the concept of quasi-atoms for IAEs in [Gd$_2$C]$^{2+}$·2e$^-$ electride. Furthermore, the quasi-atomic nature of IAEs is also confirmed by strong

localization at the center of the IAEs in the conduction electron density (CED) plotted in Fig. 3j. In addition, the CED plot shows that the IAEs also make significant contributions to the itinerant electrons. This delocalized nature of itinerant IAEs is reflected in a cylindrical Fermi surface (Supplementary Fig. 5), which is the typical characteristic of 2D electronic systems, allowing the understanding of their anisotropic transport properties of layer structured [Gd$_2$C]$^{2+}$·2e$^-$ as a 2D electride.

**Quasi-atomic IAEs with magnetic moments.** The IAEs in [Gd$_2$C]$^{2+}$·2e$^-$ electride show another distinct feature of quasi-atomic nature via their magnetic properties. The magnetization density map (MDM) in Fig. 3k shows a large magnetic moment (7.46 $\mu_B$) around Gd atoms due to unpaired electrons of the majority spin of Gd-4$f$ (Fig. 3c) and Gd-5$d$ orbitals (Fig. 3d). The IAEs form a cylindrical shape along the $c$-axis and extend

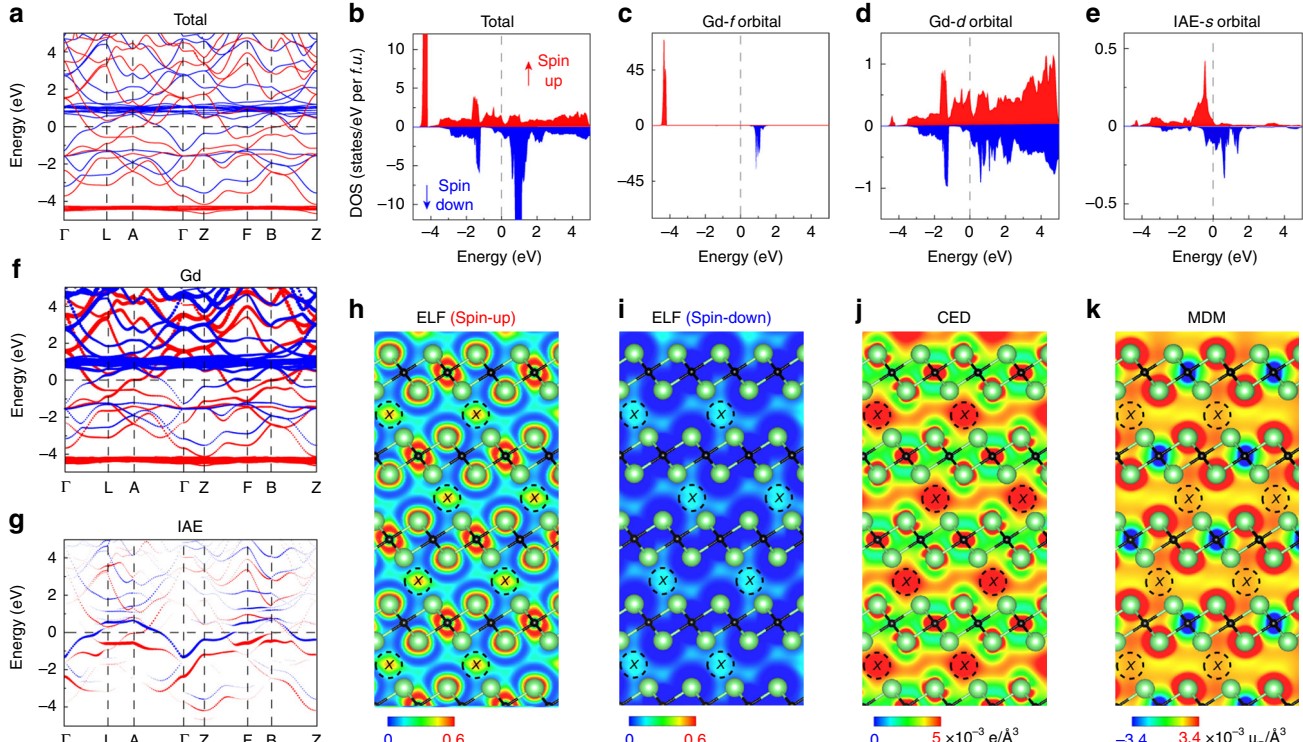

**Fig. 3 Quasi-atomic IAEs in FM [Gd$_2$C]$^{2+}$·2e$^-$ electride. a, b** Band structure and total DOS (red: spin-up; blue: spin-down). $E_F$ is set to zero energy (dashed line). **c–e** Projected DOS on Gd-$f$ orbital (**c**), Gd-$d$ orbital (**d**), and IAE-$s$ orbital (**e**). **f, g** The projected band structures on Gd atom (**f**) and IAE (**g**). **h, i** ELF for spin-up and spin-down states. "$X$" in the dashed circle denotes the site of IAEs, which is 6$c$ (0, 0, 0.5) in the $R\bar{3}m$ space group. **j** CED map (−1.0 eV < $E$ < 0 eV). **k** MDM. IAEs are strongly localized at "$X$" sites (**h, i**), contributing to a dispersion band crossing $E_F$ (**j**) and exhibiting a distinct magnetic spin density (**k**).

over the interlayer space, strongly coupling with six nearest neighbor Gd atoms in the out-of-plane direction (Fig. 3j). This feature supports the formation of the Gd–IAE–Gd linkage, which renders the characteristic MR behavior of the FM/NM system as shown in Fig. 2d, f. It is remarkable that the IAEs in [Gd$_2$C]$^{2+}$·2e$^-$ electride are FM elements with their own magnetic moment of 0.52 $\mu_B$ (See Methods for the Bader's method[33,34] and a comparison to projection sphere method in Supplementary Fig. 6, Supplementary Tables 3 and 4), which is close to the value (0.4 $\mu_B$) obtained when the 15.2 $\mu_B$ of Gd metal (two moles of gadolinium (7.6 $\mu_B$)) was subtracted from the measured 15.6 $\mu_B$ of [Gd$_2$C]$^{2+}$· 2e$^-$ electride, indicating that the enhanced magnetic moment of [Gd$_2$C]$^{2+}$·2e$^-$ electride is ascribed to the contribution from IAEs. The projected band structure for IAEs (Fig. 3g) shows both characteristics of localized and delocalized quasi-atom electrons: the IAE bands crossing the $E_F$ are delocalized and itinerant while the IAE bands slightly below the $E_F$ are mostly flat and localized. Thus, the flat and localized IAE bands are responsible for the local magnetic moments. Furthermore, our DFT calculations demonstrate that the quasi-atomic IAEs are responsible for the stronger exchange coupling in [Gd$_2$C]$^{2+}$·2e$^-$ electride than that in Gd metal. The DOS (Fig. 3d, e) and projected band structures (Fig. 3f, g) for Gd-$d$ and IAE-$s$ orbitals show that the overlap between these two orbitals is substantial, facilitating significant exchange coupling. The CED plot (Fig. 3j) reveals that the stronger exchange coupling in [Gd$_2$C]$^{2+}$·2e$^-$ electride originates from the spin-spin interaction of Gd-$d$ orbitals mediated by conduction electrons of $s$-nature IAEs, i.e., Ruderman-Kittel-Kasuya-Yosida (RKKY) interaction[35]. Consequently, this RKKY-type interaction mediated by IAEs accounts for the enhanced $T_C$ of FM [Gd$_2$C]$^{2+}$·2e$^-$ electride. However, it should be noted that

although Gd–IAE–Gd linkages in both out-of-plane and in-plane directions at interlayer space are important to impart the ferromagnetism to the [Gd$_2$C]$^{2+}$·2e$^-$ electride, the exchange coupling of Gd–C–Gd in intralayer slab is also responsible for the ferromagnetism of [Gd$_2$C]$^{2+}$·2e$^-$ electride, indicating that the ferromagnetism occurs in 3D.

**Ferromagnetic IAEs.** The FM quasi-atomic nature of IAEs in [Gd$_2$C]$^{2+}$·2e$^-$ electride was further verified by comparing the experimental characterizations of Cl-substituted [Gd$_2$C]$^{2+}$·(1 − $x$)2e$^-$·Cl$_x$ system and theoretical calculations of IAE-removed [Gd$_2$C]$^{2+}$·$y$□·(2 − $y$)e$^-$ system (□ represents the vacancy of IAE) that cannot be synthesized in experiments. When IAEs are removed from [Gd$_2$C]$^{2+}$·2e$^-$ electride, an AFM transition occurs: from strong FM [Gd$_2$C]$^{2+}$·2e$^-$ electride ($T_C$ ~ 393 K) to weaker FM [Gd$_2$C]$^{2+}$·1□·1e$^-$ ($T_C$ ~ 241 K) and eventually to AFM [Gd$_2$C]$^{2+}$·2□ (Néel temperature, $T_N$ ~ 58 K). This magnetic phase transition corresponds to a change from the FM spin alignment of Gd atoms in [Gd$_2$C]$^{2+}$·2e$^-$ and [Gd$_2$C]$^{2+}$·1□·1e$^-$ to the AFM alignment of Gd atoms in [Gd$_2$C]$^{2+}$·2□ (Supplementary Fig. 7). Also, we calculated a hypothetical Gd lattice by making a structure of Gd atoms in the same position of [Gd$_2$C]$^{2+}$·2e$^-$ electride without C atoms. However, the hypothetical structure is energetically unstable with a much higher formation energy (+0.105 eV per Gd atom compared to −0.382 eV per Gd atom of [Gd$_2$C]$^{2+}$·2e$^-$), indicating that IAEs are critical for constructing the layer structured ionic crystal and imparting the FM properties of [Gd$_2$C]$^{2+}$·2e$^-$ electride. This result suggests that the strongly localized IAEs facilitate the FM spin alignment of Gd atoms in [Gd$_2$C]$^{2+}$·2e$^-$ and [Gd$_2$C]$^{2+}$· 1□·1e$^-$ electrides. We thus conclude that the IAEs are inherent

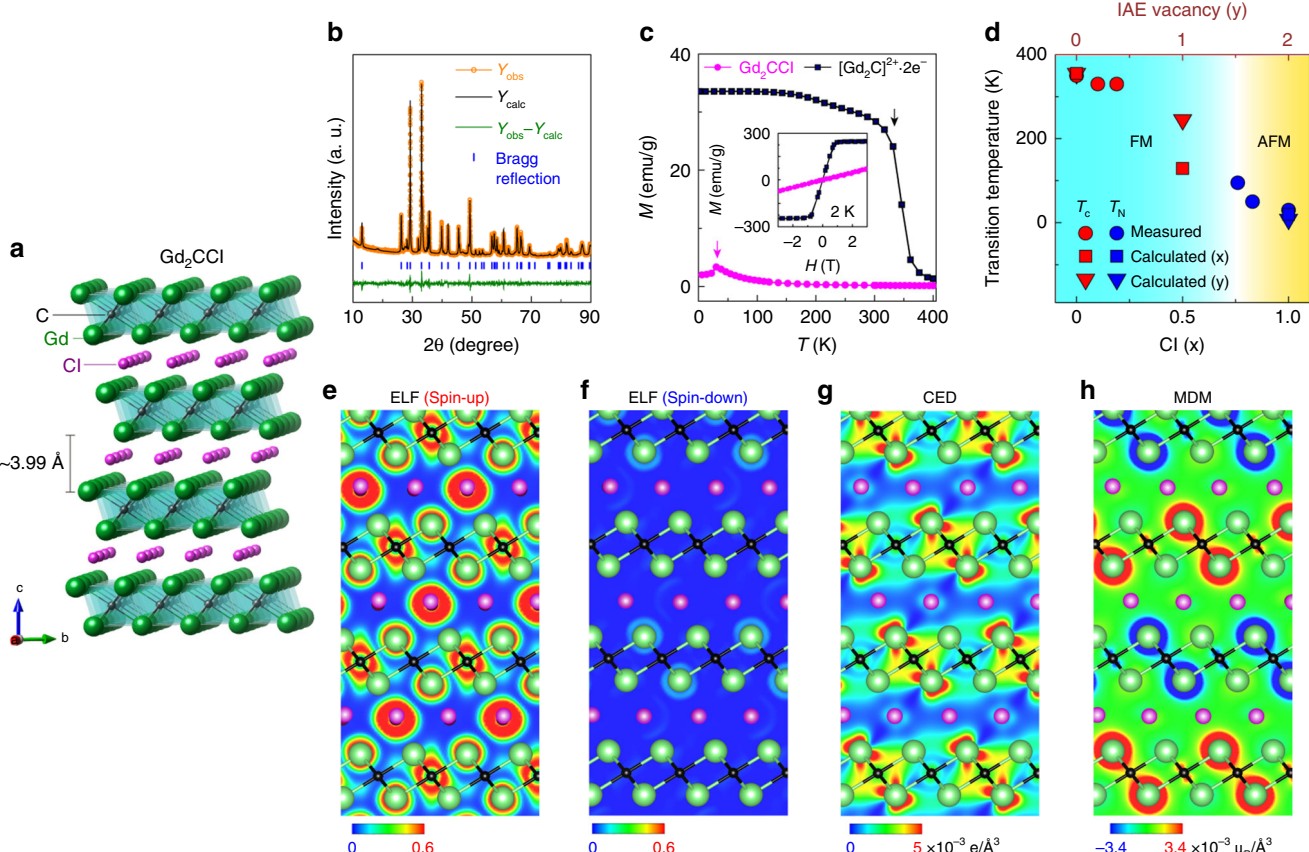

**Fig. 4 FM quasi-atomic IAEs in $[Gd_2C]^{2+}\cdot2e^-$ electride. a** Schematic illustration of the crystal structure of $Gd_2CCl$ containing Cl atoms substituted for IAEs. **b** Rietveld refinement of powder XRD patterns for $Gd_2CCl$. **c** Temperature ($T$) dependence of magnetization ($M$) for polycrystalline $[Gd_2C]^{2+}\cdot2e^-$ electride (black) and $Gd_2CCl$ (pink) under a magnetic field of 0.1 T. Each arrow indicates the $T_C$ (350 K, black) and $T_N$ (30 K, pink). The inset is the $M$–$H$ curve for both samples at 2 K. **d** Magnetic phase diagram for $[Gd_2C]^{2+}\cdot(1-x)2e^-\cdot Cl_x$ (bottom scale) and $[Gd_2C]^{2+}\cdot y\square\cdot(2-y)e^-$ (top scale). **e, f** ELF for spin-up (**e**) and spin-down (**f**) states for $Gd_2CCl$. **g, h** CED map ($-1.0\,eV < E < 0\,eV$) (**g**) and MDM (**h**) for $Gd_2CCl$.

FM quasi-atoms responsible for the transition from the AFM $[Gd_2C]^{2+}$ lattice framework to the FM $[Gd_2C]^{2+}\cdot2e^-$.

Finally, to verify the FM nature of quasi-atomic IAEs in the $[Gd_2C]^{2+}\cdot2e^-$ electride, we experimentally removed IAEs by the substitution of paramagnetic Cl atoms into $[Gd_2C]^{2+}\cdot2e^-$ electride to generate the $[Gd_2C]^{2+}\cdot(1-x)2e^-\cdot Cl_x$ system. The substituted Cl atoms were found to occupy the same Wyckoff positions of quasi-atomic IAEs and to bond ionically with neighboring Gd atoms (Fig. 4a, b; Supplementary Fig. 8). The temperature dependence of the magnetic moment (Fig. 4c) clearly indicates that non-electride $Gd_2CCl$ is an AFM system with a $T_N$ of 30 K, which is close to the calculated $T_N$ values (Fig. 4d) for both $Gd_2CCl$ (22 K) and the IAE-removed $[Gd_2C]^{2+}$ lattice framework (58 K). Figure 4e–h shows the results of DFT calculations for the $Gd_2CCl$. Furthermore, the MDM of $Gd_2CCl$ is nearly identical to that of $[Gd_2C]^{2+}\cdot2\square$, showing that Cl atoms at the position of IAEs have no magnetic moment and Gd atoms across Cl atoms have an AFM spin alignment (Fig. 4h). This magnetic phase transition is explained by the change in the exchange interaction ($J$) between Gd atoms mediated by IAEs. The exchange interaction obtained from DFT calculations shows that $J_1$ (between in-plane Gd–Gd atoms across IAEs or Cl atoms) has a smaller magnitude than $J_2$ (between out-of-plane Gd–Gd atoms across IAEs or Cl atoms) in all cases considered here (Supplementary Figs. 9 and 10). The comparison of MDMs between $[Gd_2C]^{2+}\cdot2e^-$ electride and $Gd_2CCl$ (Figs. 3k and 4h) clearly demonstrates that the polarized spins of Gd atoms for $J_2$ are more strongly coupled than those for $J_1$ when the IAEs occupy

the "$X$" site. Moreover, $J_1$ and $J_2$ are simultaneously attenuated as the IAEs are substituted by Cl atoms, causing a decrease in $T_C$ with an increase of $x$ in $[Gd_2C]^{2+}\cdot(1-x)2e^-\cdot Cl_x$ (Fig. 4d; Supplementary Figs. 9 and 11). However, when "$X$" exceeds the critical concentration of 0.75, the IAEs interacting with Gd atoms become dilute, and the $f$-electrons of Gd atoms preferentially adopt an antibonding character; this effect diminishes the exchange interaction between Gd atoms in the intralayer slab, leading to a stable AFM spin alignment. These results indicate that the quasi-atomic IAEs initiate the exchange interaction of out-of-plane Gd–Gd atoms across the IAEs and that a subsequent exchange interaction of in-plane Gd–Gd atoms occurs, facilitating FM ordering in $[Gd_2C]^{2+}\cdot2e^-$ electride. Besides the exchange interactions of IAEs mediated Gd atoms in the interlayer space, the exchange interaction of out-of-plane Gd–Gd atoms across C atoms ($J_3$) in the intralayer slab also contributes to the ferromagnetism of $[Gd_2C]^{2+}\cdot2e^-$ electride, mainly affecting the out-of-plane magnetic properties. Thus, we conclude that the ferromagnetism of layer structured 2D $[Gd_2C]^{2+}\cdot2e^-$ electride occurs in 3D with strong anisotropic characters.

## Discussion

To conclude, our work provides a new understanding on the magnetism of IAEs, which behave as FM quasi-atoms with their own magnetic moment. This phenomenon is realized in a 2D electride consisting of strongly localized IAEs occupying specific atomic sites in interlayer space. The quasi-atomic IAEs with their own magnetic moment in 2D $[Gd_2C]^{2+}\cdot2e^-$ electride play an

important role as magnetic elements introduced in FM alloys to enhance the magnetic properties of the material[28]. Furthermore, our findings open a new possibility that the quasi-atomic IAE can be a promising ingredient to develop FM electrides consisting only paramagnetic elements or small amounts of rare-earth elements.

## Methods

**Sample preparation and single-crystal growth.** All manipulations were carried out in glove boxes filled with recirculating high-purity Ar (99.999%) to suppress oxygen and moisture concentrations to less than 0.1 ppm because $[Gd_2C]^{2+} \cdot 2e^-$ and $[Gd_2C]^{2+} \cdot (1-x)2e^- \cdot Cl_x$ are highly reactive under ambient conditions. For single-crystal growth using the floating zone (FZ) melting method, stoichiometric polycrystalline $[Gd_2C]^{2+} \cdot 2e^-$ rods were synthesized by the arc melting method to prepare the feed and seed materials. We mixed Gd metal pieces and graphite pieces in a 2:1 molar ratio and melted the mixture under a high-purity argon atmosphere in arc furnaces. To obtain a single-phase and ensure the homogeneity of the polycrystalline $[Gd_2C]^{2+} \cdot 2e^-$, we repeated the melting process at least three times. After cooling, the polycrystalline $[Gd_2C]^{2+} \cdot 2e^-$ ingot was shaped into a rod in the glove boxes. The FZ melting method was executed under a high-purity argon atmosphere (Ar = 99.999 %) to prevent oxidation. The feed and seed rods were rotated in opposite directions at the same speed of 6 rpm, with the low-melt viscosity of the $[Gd_2C]^{2+} \cdot 2e^-$ electride causing a growth speed slower than 2 mm per hour. The grown single-crystal $[Gd_2C]^{2+} \cdot 2e^-$ electride was applied to the measurements of structural, electrical, magnetic and magnetotransport properties. $[Gd_2C]^{2+} \cdot (1-x)2e^- \cdot Cl_x$ samples were synthesized by the solid-state reaction method. We prepared a pellet of mixed powders of a pulverized polycrystalline $[Gd_2C]^{2+} \cdot 2e^-$ ingot, $GdCl_3$ (99.99%), and graphite. Then, the pellet was wrapped with molybdenum foil and sealed in a quartz tube under $10^{-3}$ Pa. The vacuum-sealed quartz tube containing the pellet was annealed at 1100 °C for 96 hrs. in a box furnace. We have checked the contents of impurities for Gd raw metal pieces and FZ grown $[Gd_2C]^{2+} \cdot 2e^-$ single-crystals by the inductively coupled plasma (ICP) spectroscopy. The ICP results shown in Supplementary Table 5 verifies that the concentrations of all inspected impurities (except Tb and Er ferromagnets with around 1 ppm) are below 1 ppm.

**Characterization of electrical and magnetic properties.** Sample and device preparation for characterization of physical properties were carried out in glove boxes filled with recirculating high-purity Ar (99.999%). To measure the electrical properties of single-crystal $[Gd_2C]^{2+} \cdot 2e^-$ electride, electrical contacts in the four-point probe configuration were created with silver epoxy on a cleaved surface. After the contacts were made, Apiezon N grease was coated onto the sample surfaces to prevent oxidation during measurements. For the measurement of MR and the Hall effect, we adopted the stamp method to prohibit the lifting of FM samples by the applied magnetic field. Single-crystal $[Gd_2C]^{2+} \cdot 2e^-$ electride was cleaved by 3 M Scotch tape, and the exfoliated crystal was pressed with 3 M Scotch tape onto the patterned electrodes on a $SiO_2/Si$ wafer. The transferred sample was then pressed with a copper plate and firmly fixed. For the measurement of magnetic properties using a vibrating sample magnetometer, a plastic capsule copula containing a weighted sample was coated with N grease to prevent the oxidation of samples. We calculated magnetic moments based on the values of saturation magnetization ($M_S$) obtained at 2 K for single-crystal $[Gd_2C]^{2+} \cdot 2e^-$ electride and Gd metal. The magnetic moment is given by $M_S/N \cdot \mu_B$, where N is the number of elements, and $\mu_B$ is the Bohr magneton constant.

**XAS and XMCD.** XAS and XMCD measurements were carried out at the 2A beamline of the Pohang Accelerator Laboratory. Cleaved single-crystal $[Gd_2C]^{2+} \cdot 2e^-$ electride was attached to a copper holder with a Torr seal and contacted with silver epoxy. To carry and install the sample into the measurement chamber, the sample was attached to a copper holder and placed in water-free hexane liquid, which would not react with the sample, to prevent oxidation. After the sample was placed in the chamber, the hexane liquid covering the sample was vaporized by heating the chamber to ~100 °C for 12 hrs. The measurements were conducted at 20 K under an ultra-high vacuum (~$10^{-10}$ Torr) with an applied magnetic field of ~0.7 T along the cleaved surface. XAS and XMCD measurements were carried out at Gd $M_{4,5}$ ($3d_{5/2} \rightarrow 4f_{7/2}$, $3d_{3/2} \rightarrow 4f_{5/2}$ transitions) and C K-edge, which were compared with those of Gd metal and graphite.

**Electronic and magnetic structure calculations.** All ab initio total-energy calculations and geometry optimizations were performed within DFT using the generalized gradient approximation (GGA) with the Perdew–Burke–Ernzerhof functional and the projected augmented wave method, as implemented in Vienna ab initio simulation package[36–38]. A primitive rhombohedral unit cell containing one chemical formula was used for FM spin configurations, while a quadruple-size unit cell was used for AFM configurations. The electron wave functions were expanded in a plane-wave basis set with a cutoff energy of 520 eV. The Brillouin zone was sampled using a $108 \times 108 \times 96$ Monkhorst–Pack k-point set for ELF, partial charge density, and magnetization density calculations. All calculations were

spin-polarized, and the positions of atoms and the size and shape of the unit cell were fully relaxed to obtain the optimized lattice structure. An empty sphere with a Wigner-Seitz radius of 1.25 Å was used to obtain the projected DOS on the interstitial position ("X" site). The local magnetic moment of each ion including the IAE was computed by extending Bader's charge decomposition method to magnetization densities that can attain negative values. The Bader basin for each site is computed as the volume containing a single magnetization density maximum and is separated from other volumes by a zero-flux surface of the gradients of the magnitude of the magnetization density. Once a Bader basin is determined, the atomic magnetic moment is computed by integrating the signed value of magnetization density within the basin. Our calculation shows that this new approach of applying Bader analysis to magnetization density provides more accurate and robust measure of magnetic moment for a system with delocalized IAEs such as $[Gd_2C]^{2+} \cdot 2e^-$ electride than the conventional projection method that strongly depends on the size of artificially set projection spheres and inevitably suffers the problem of undercounting and double counting (see the comparison in Supplementary Fig. 6; Supplementary Tables 3 and 4). Spin configurations used for the calculation of $J$ in Supplementary Fig. 9 for $[Gd_2C]^{2+} \cdot (1-x)2e^- \cdot Cl_x$ system viewed from $(11\bar{2})$ direction. Given the Heisenberg Hamiltonian for magnetic energy, $H_M = -\sum_{ij} \hat{e}_i \cdot J_{ij} \cdot \hat{e}_j$, ($\hat{e}_i$ is the unit vector in the direction of the $i$th site magnetization $J_{ij}$ are the exchange parameters), these eight configurations are used to determine the exchange parameters $J_{ij}$ (Supplementary Fig. 10), which in turn determine magnetic critical temperatures $T_C$ and $T_N$.

## Data availability

The authors declare that the main data supporting the findings of this study we contained within the paper. All other relevant data are available from the corresponding author upon reasonable request.

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

## Acknowledgements

This work was supported by a National Research Foundation of Korea (NRF) grant funded by the Korean government (Ministry of Science, ICT & Future Planning) (No. 2015M3D1A1070639) and in part by the Center for Computational Sciences (CCS) at Mississippi State University. Computer time allocation was provided by the High Performance Computing Collaboratory (HPC²) at Mississippi State University. We acknowledge J. Lee and D. Kim for the measurement of Hall coefficient at low temperatures.

## Author contributions

S.W.K. conceived the idea and organized the research. S.Y.L. and J.P. grew the single-crystals and fabricated all polycrystals. C.N.N., S.G.K., and S.W.K. carried out the computational studies. S.Y.L. performed the transport and magnetic measurements. S.Y.L., J.Y.H., S.G.K., and S.W.K. analyzed the structural, electrical and magnetic properties. S.Y.L. and Y.H.K. performed the XAS and XMCD measurements. S.G.K., Y.Z., Y.M., H.H., and S.W.K. analyzed the magnetic properties. J.B., K.L., K.H.L., and Y.H.L. discussed the results and commented on the paer. All of the authors co-wrote the paper.

## Competing interests

The authors declare no competing interests.
