## [Peer Review File · Nature Communications]

Reviewers' Comments:

Reviewer #1:

Remarks to the Author:

This paper shows very interesting experiment and computation results on a two-dimensional electrified Gd₂C. Basically, it demonstrates that such system can give rise to ferromagnetic coupling of the local spins mediated by the anionic electrons occupying the quasi-atom s-like orbitals. This is a new and interesting progress in the area of electrified compounds as well as 2D materials. It also provides deeper understanding of the bonding features and functions of quasi-atoms. I believe this paper will eventually become a very interesting and important contribution to Nature Communication. The analysis and the presentation of the electronic structures and the ferromagnetic mechanism do need sufficient improvement so to solidify the conclusions and the understanding of this fascinating material. Here are my detailed comments.

1) Are there local magnetic moments locating on the quasi-atoms? The experimental value of 0.4 μ_B was obtained by subtracting magnetic moment of Gd from Gd₂C, which is not accurate. The DFT calculations strongly rely on the projection of wave functions into atomic orbitals, which will strongly depend on the projection sphere. Because the s orbital and the Gd d orbitals are strongly coupled, the projection may actually have large quantity of Gd d. There is one way that can systematically check the origin of the local moments. One can try to calculate the projected local moments by gradually reducing the sphere radius of quasi-atoms and the Gd atoms, and see how the local moments change with it.

Although it will be exciting to see that the electrons locating on the s-like quasi-atom orbitals being spin polarized and contribute to the total magnetic moments, it is not crucial for the importance of this work. Overall, the magnetic moment from quasi-atom, even if it is true, is only a small part. The important function of these s-like electrons is to mediate the ferromagnetic interactions among the local moments of the Gd atoms.

2) Are the quasi-atom electrons localized? The paper wanders between a picture of localized quasi-atom electrons with significant spin polarization and itinerant electrons that can mediate the ferromagnetic interactions of Gd local moments. The truth might lie in between the two. Based on the results shown in the paper, the latter is a better approximation. The quasi-atom electrons are more itinerant and their insignificant spin polarization is induced by their coupling with the local moments. The corresponding bands are not really flat as claimed in the paper. The localization of these electrons should strongly depend on the interatomic distances especially between the neighboring quasi-atoms. It will be helpful to compare that with those in Ca₂N and Y₂C. The paper shows that the inter-layer distances are shorter for Gd₂Y, which does not necessarily indicate the localization of the electrons on quasi-atoms.

3) Several minor points: a) It is not quite right to say "An electrified is a generalized form of interstitial anionic electrons (IAEs) confined in positively charged cavities." The cavity is not necessarily to be charged. b) line 97, I don't see correlation between the ferromagnetic electrified and 2D Wigner crystal. The latter is a competing result between the kinetic energy and Coulomb potential energy. c) line 260, I don't think IAE will be a best strategy for the development of permanent magnets, they should still rely more on the f metals. d) The stretch of current work to fundamental physics of Wigner crystal is too far-reaching.

Reviewer #2:

Remarks to the Author:

After reading through this paper, I did not find strong evidence to support what the authors claimed.

First, this is not a "two-dimensional" system. The author also noticed that the ferromagnetic coupling, if it indeed exists as the author claimed, is both interlayer and intralayer. This means "3D" rather than "2D".

Second, assume it was 2D, how come the in-plane anisotropy could result in such a high temperature 2D magnetic order?

Third, authors also noticed (see fig. 2b,c) metal Gd also exhibits ferromagnetism. So, in their compound, is it possible the observed magnetic signal is from Gd lattice (in compound) rather than from the anionic electrons? After all, such compound is like a strained Gd or "decorated" Gd. The authors need to rule out this possible scenario based on solid evidence. This point directly relates to whether the author can claim ferromagnetic electrone or not.

At last, although evidences like XRD shows the quality of the crystal, we know XRD has detection limit and cannot rule out all impurity phase or defects especially in low concentration (e.g., <0.1%). But throwing the bulk crystals as a whole into magnetometers for measurements may not be careful enough: even <0.1% foreign species or phase can give appreciable magnetic signals given the large chunk of samples. In fact, this is a major source of artifacts that have been routinely measured days after days in worldwide labs. The authors need solid evidence to rule out this scenario, if they insist their claims.

Reviewer #3:

None

Response Letter

Reviewer #1

Comment:

This paper shows very interesting experiment and computation results on a two-dimensional electride Gd_2C . Basically, it demonstrates that such system can give rise to ferromagnetic coupling of the local spins mediated by the anionic electrons occupying the quasi-atom s-like orbitals. This is a new and interesting progress in the area of electride compounds as well as 2D materials. It also provide deeper understanding of the bonding features and functions of quasi-atoms. I believe this paper will eventually become a very interesting and important contribution to Nature Communication. The analysis and the presentation of the electronic structures and the ferromagnetic mechanism do need sufficient improvement so to solidify the conclusions and the understanding of this fascinating material. Here are my detailed comments.

Ans.) We greatly appreciate your positive evaluations. We do concur with your insight that the ferromagnetic coupling of the local spins mediated by the anionic electrons acting as quasi-atoms will be a new interesting phenomenon in 2D materials beyond electrides. Following your valuable comments, we have verified your suggestions and revised the manuscript to provide a concrete evidence for the ferromagnetism of $[\text{Gd}_2\text{C}]^{2+}\cdot 2\text{e}^-$ electride.

1. Are there local magnetic moments locating on the quasi-atoms? The experimental value of $0.4 \mu_B$ was obtained by subtracting magnetic moment of Gd from Gd_2C , which is not accurate. The DFT calculations strongly rely on the projection of wave functions into atomic orbitals, which will strongly depends on the projection sphere. Because the s orbital and the Gd d orbitals are strongly coupled, the projection may actually have large quantity of Gd d. There is one way that can systematically check the origin of the local moments. One can try to calculate the projected local moments by gradually reducing the sphere radius of quasi-atoms and the Gd atoms, and see how the local moments change with it.

Ans. 1) Thank you very much for these valuable comments that can help us to strengthen our main claim. Yes, we assert that local magnetic moments are indeed located on the quasi-atoms. We believe the reviewer is referring to the statement of experimental value of $0.4 \mu_B$ in line 201 in the

revised manuscript. We would like to note that we do not claim the value of $0.4 \mu_B$ as a conclusive experimentally measured value for the local magnetic moment of the quasi-atom of interstitial anionic electrons (IAEs) in the $[\text{Gd}_2\text{C}]^{2+}\cdot 2\text{e}^-$ electrified. Rather, we are making a qualitative argument to point out the fact that our DFT calculated value of $0.52 \mu_B$ is quite close to the difference between the corresponding values for $[\text{Gd}_2\text{C}]^{2+}\cdot 2\text{e}^-$ electrified and Gd metal leading to the conclusion that the enhanced magnetic moment can be ascribed to the contribution from IAEs. Thus, in order to further demonstrate the distinct contribution from IAEs, we calculated the magnetic moments of positively charged lattice framework $[\text{Gd}_2\text{C}]^{2+}\cdot 2\Box$ without interstitial anionic electrons (IAEs), as listed in Extended Data Table 2. Our calculation showed that the magnetic moment of $[\text{Gd}_2\text{C}]^{2+}\cdot 2\Box$ in the absence of IAEs is $14.2 \mu_B$ that is much smaller than the value of $15.3 \mu_B$ (DFT) of $[\text{Gd}_2\text{C}]^{2+}\cdot 2\text{e}^-$ electrified, in which the magnetic moments of IAE are $0.52 \mu_B$.

Regarding the magnetic moment of the quasi-atoms, we would like to note that our DFT calculated values for the local magnetic moment of the constituent atoms and quasi-atomic IAEs are not obtained from the projection on spheres with adjustable Wigner-Seitz radii. Instead, as we described in the Method section for DFT calculation, we calculate the local magnetic moments using the Bader method [Ref 01] where the interstitial space is divided up into atomic volumes where the dividing surfaces are at a minimum in the magnetization density, i.e. the gradient of the magnetization density is zero along the surface normal. Therefore, the value of $0.52 \mu_B$ for the local magnetic moment of the quasi-atomic IAE is independent of the radius of the projection sphere.

Figure R1 below shows how the volume of the unit cell is assigned to each atom and IAEs. In Bader method (Fig. R1a), the entire volume is divided according to the profile of magnetization density without any unassigned or double counted volume. In comparison, if we use the projection to spheres centered on each atom (Fig. R1b) to compute the local magnetic moment, some magnetization density will be double-counted (red region) and some region (especially the interstitial space) will be unaccounted for (blue region).

In Table R1 below, we compare the Bader method and the projection method to compute the local magnetic moments. Because in Bader method, all volumes are counted without double-counting or under-counting, the sum of local magnetic moments are exactly equal to the total magnetic moment computed from the integration of magnetization density over the unit cell volume. In comparison, the projection method cannot avoid over-counting or under-counting of some regions. Consequently, it leads to inaccurate valuation of local magnetic moments and their sum does not match with the total magnetic moment.

Figure R1. Bader basin and projection spheres of $[\text{Gd}_2\text{C}]^{2+}\cdot 2\text{e}^-$ electride. **a**, Bader basin for magnetization density of $[\text{Gd}_2\text{C}]^{2+}\cdot 2\text{e}^-$. Each color represents the volume assigned to different atom (blue: Gd1, green: Gd2, yellow: C, red: IAEs). There is no volume unaccounted and no overlapped volume. **b**, The projection spheres assigned to each atom when projection method is employed. Each color represents the number of spheres the point is assigned to (blue: 0, green: 1, red: 2). The volumes of spheres cover 92.5% of the total volume. There are unaccounted (blue) and double counted (red) regions.

To address the suggestion the reviewer made, we computed the local magnetic moments using the projection method as we gradually change the radii of the projection spheres. Table R2 below shows the result. When we reduce the radii of projection spheres of Gd atom and IAEs by 8.3% and 16.7%, we see very small change in the local magnetic moment of Gd atom (-1.22% and -3.13%). On the other hand, the local magnetic moment of IAEs change significantly by -21.9% and -40.6%. This indicates that the local magnetic moment of Gd atom is well localized about the nucleus and suffer little reduction when the radius of the projection sphere is reduced. It also means that Gd-d orbital does not contribute much to the magnetic moment of IAEs. On the other hand, the magnetization density of IAEs is significantly delocalized over the interstitial region and its local magnetic moment is reduced when a smaller projection sphere is employed.

These results obtained from projection method are newly added in the Supplementary Information (Fig. 9, Table 3 and Table 4) to clearly demonstrate the crucial contribution of IAEs for the total magnetic moment and to explain the contribution of each ferromagnetic species with the projection sphere radii dependence of local magnetic moment.

Table R1. Comparison of Bader and projection method to compute local magnetic moment. The “sum” column is computed by adding the local moments of all atoms and IAEs. The “total” column is the total magnetic moment obtained by integrating the magnetization density over the entire unit cell.

Method	Mag. mom. (μ_B)					Vol %
	Gd	C	IAE	Sum	Total	
Bader	7.46	-0.12	0.52	15.33	15.33	100%
Projection	7.38	-0.11	0.32	14.97	15.33	92.5%

Table R2. Local magnetic moments on Gd atom and IAEs as the radii of projection spheres are changed. $\Delta r/r$ and $\Delta\mu/\mu$ represent the percentage of change in sphere radius and local magnetic moment, respectively.

	Gd			IAE		
	r (Å)	1.778	-1.630	1.482	1.499	1.374
$\Delta r/r$	0.0	-8.3 %	-16.7 %	--	-8.3 %	-16.7 %
μ (μ_B)	7.38	7.27	7.13	0.32	0.25	0.19
$\Delta\mu/\mu$	---	-1.22 %	-3.13 %	---	-21.9 %	-40.6 %

2. Are the quasi-atom electrons localized? The paper wanders between a picture of localized quasi-atom electrons with significant spin polarization and itinerant electrons that can mediate the ferromagnetic interactions of Gd local moments. The truth might lie in between the two. Based on the results shown in the paper, the latter is a better approximation. The quasi-atom electrons are more itinerant and their insignificant spin polarization is induced by their coupling with the local moments. The corresponding bands are not really flat as claimed in the paper. The localization of these electrons should strongly depend on the interatomic distances especially between the neighboring quasi-atoms. It will be helpful to compare that with those in $[\text{Ca}_2\text{N}]^+ \cdot e^-$ and $[\text{Y}_2\text{C}]^{2+} \cdot 2e^-$. The papers shows that the inter-layer distances are shorter for $[\text{Gd}_2\text{C}]^{2+} \cdot 2e^-$, which does not necessary indicate the localization of the electrons on quasi-atoms.

Ans. 2) We agree with the reviewer that the quasi-atom electrons possess both localized and itinerant characteristics. The electron localization function (ELF) (Fig. 3h and 3i) and magnetization density (Fig. 3k) show a strong localization of the IAEs. On the other hand, the conduction charge density (Fig. 3j) shows that the IAEs make significant contribution to the itinerant electrons. In addition, projected band structure for IAEs (Fig. 3g) shows both characteristics of quasi-atom electrons: the IAE bands crossing the Fermi level are delocalized and itinerant while the IAE bands slightly below the Fermi level are mostly flat and localized. Thus, flat and localized IAE bands are responsible for the local magnetic moments. **The corresponding sentences are added in the revised manuscript.**

On the other hand, we could not establish a relationship between the distances of neighboring quasi-atomic IAEs and the degree of localization as shown in Fig. R2. The interatomic distances of quasi-atomic IAEs in three two-dimensional electrides are nearly identical. However, by comparing with a small maxima of IAEs in a rather larger interlayer spacing (3.86 Å) of [Ca₂N]⁺·e⁻, we can infer that the strong localization of IAEs of [Y₂C]²⁺·2e⁻ and [Gd₂C]²⁺·2e⁻ is related to their smaller interlayer spacing (3.29 Å and 3.38 Å). Nevertheless, as the reviewer noted, the quantitative theoretical analysis on the degree of localization of IAEs will be a very interesting topic, if combined with the experimental study such as pressure effect as we have examined [Ref 02].

From the consideration of this comments, our results indicates the importance of constituent elements of electrides for ferromagnetism, such as Gd of $[\text{Gd}_2\text{C}]^{2+}\cdot 2\text{e}^-$, in which IAEs are responsible to impart ferromagnetic properties into $[\text{Gd}_2\text{C}]^{2+}\cdot 2\text{e}^-$ electrides due to the strong exchange interaction with magnetic Gd f-metal. This makes it possible to understand why the $[\text{Y}_2\text{C}]^{2+}\cdot 2\text{e}^-$ does not show the ferromagnetism. Thank you again for these constructive comments.

3.

a) It is not quite right to say "An electride is a generalized form of interstitial anionic electrons (IAEs) confined in positively charged cavities." The cavity is not necessarily to be charged.

Ans. 3) We fully agree with the reviewer. We had a mistake in composing the statement. We revised the statement to read "An electride is a generalized form of cavity-trapped interstitial anionic electrons (IAEs) in a positively charged lattice framework"

b) line 97, I don't see correlation between the ferromagnetic electride and 2D Wigner crystal. The later is a competing result between the kinetic energy and Coulomb potential energy

Ans. 4) We agree with the reviewer that we didn't present a link between the ferromagnetic electride and 2D Wigner crystal in the present work. Therefore, we revised the statement by removing the related term.

c) line 260, I don't thing IAE will be a best strategy for the development of permanent magnets, they should still rely more on the f metals.

Ans. 5) Thank you for the constructive comment. We respectfully agree with reviewer's opinion. As discussed, the f metal, such as Gd in $[\text{Gd}_2\text{C}]^{2+}\cdot 2\text{e}^-$ electride, is essential for the ferromagnetism of electride, if only the scientific findings of the present work are considered. Nevertheless, reviewing studies with other electride systems, which are composed of nonmagnetic elements, also shows ferromagnetism without f metals. Thus, in expectation of the future of magnetic electride, we used the term permanent magnets. However, to avoid unnecessary misunderstanding, we revised the statement by removing the related term.

d) The stretch of current work to fundamental physics of Wigner crystal is too far-reaching.

Ans. 6) We agree with the reviewer that we didn't present experimental evidences to demonstrate the formation of a 2D Wigner crystal in the present work. We revised the statement to remove the disputed term. However, we have obtained the experimental evidence of the Wigner crystal in the two-dimensional electride, thus it will be reported in a separate paper elsewhere.

References

[Ref 01] Bader, R. F. W. A quantum theory of molecular structure and its applications. *Chem. Rev.* **91**, 893-928 (1991).

[Ref 02] J. Park, et al. , Tuning the Spin-Alignment of Interstitial Electrons in Two-Dimensional Y₂C Electride via Chemical Pressure, *J. Am. Chem. Soc.*, 2017, **139** (48), pp 17277–17280 (2017).

Reviewer #2

1. After reading through this paper, I did not find strong evidence to support what the authors claimed. First, this is not a “two-dimensional” system. The author also noticed that the ferromagnetic coupling, if it indeed exists as the author claimed, is both interlayer and intralayer. This means “3D” rather than “2D”.

Ans. 1) Thank you for the constructive comment. We concur that the reviewer is making a valid argument. We believe the $[\text{Gd}_2\text{C}]^{2+}\cdot 2\text{e}^-$ electride should be considered a two-dimensional system in terms of structure and electron transport while we agree with the reviewer that the $[\text{Gd}_2\text{C}]^{2+}\cdot 2\text{e}^-$ electride does have a “3D” character in terms of the magnetic interaction. First, in terms of the structure, $[\text{Gd}_2\text{C}]^{2+}\cdot 2\text{e}^-$ crystal is definitely two-dimensional. As shown in Fig 1a of the manuscript, $[\text{Gd}_2\text{C}]^{2+}\cdot 2\text{e}^-$ crystal is composed of $[\text{Gd}_2\text{C}]^{2+}$ layer units formed by edge-sharing Gd_6C octahedrals. These two-dimensional layers are separated by 3.38 Å gap along the *c*-axis. This distance is larger than the thickness of the $[\text{Gd}_2\text{C}]^{2+}$ layer (2.76 Å) (see Fig. 3 of this letter) and it asserts that the structure of $[\text{Gd}_2\text{C}]^{2+}\cdot 2\text{e}^-$ crystal is two-dimensional. Secondly, $[\text{Gd}_2\text{C}]^{2+}\cdot 2\text{e}^-$ shows strong two-dimensional characters in terms of electron transport. As shown in Fig 2a, the measured electrical resistivity in the in-plane direction is ~60 times smaller than that of the out-of-plane direction. We also calculated the Fermi surfaces of the $[\text{Gd}_2\text{C}]^{2+}\cdot 2\text{e}^-$ crystal (Fig. R3 shown below and added as Supplementary Fig 6 in the revised manuscript) and they show predominantly two-dimensional characteristics as the electron velocity is normal to the constant energy surface in *k*-space. And we agree with the reviewer that in terms of magnetic interaction, $[\text{Gd}_2\text{C}]^{2+}\cdot 2\text{e}^-$ electride does have a 3D character as it has interlayer exchange interaction across the IAE layer. To reflect this point, we carefully revised our manuscript to make sure that we don't refer $[\text{Gd}_2\text{C}]^{2+}\cdot 2\text{e}^-$ electride as a 2D magnetic system.

Figure R3. Fermi surfaces of $[\text{Gd}_2\text{C}]^{2+}\cdot 2\text{e}^-$ electride, a–b Spin-up (a) and spin-down (b) Fermi surfaces. c High symmetry special k-points of the reciprocal lattice.

2. Assume it was 2D, how come the in-plane anisotropy could result in such a high temperature 2D magnetic order?

Ans. 2) Thank you for the careful comment. As described in our reply to the previous comment, we are not claiming that $[\text{Gd}_2\text{C}]^{2+}\cdot 2\text{e}^-$ electride has 2D magnetic order. To reflect this point, we carefully revised our manuscript to make sure that we don't refer $[\text{Gd}_2\text{C}]^{2+}\cdot 2\text{e}^-$ electride as a 2D magnetic system.

3. Authors also noticed (see fig. 2b,c) metal Gd also exhibits ferromagnetism. So, in their compound, is it possible the observed magnetic signal is from Gd lattice (in compound) rather than from the anionic electrons? After all, such compound is like a strained Gd or “decorated” Gd. The authors need to rule out this possible scenario based on solid evidence. This point directly relates to whether the author can claim ferromagnetic electride or not.

Ans. 3) Thank you for comment. We agree with the reviewer's point that the main contribution of the ferromagnetic ordering of $[\text{Gd}_2\text{C}]^{2+}\cdot 2\text{e}^-$ comes from Gd lattice as Gd is such a strong ferromagnetic material. However, the evolution of ferromagnetism and the source of enhanced Curie temperature and magnetic moment of $[\text{Gd}_2\text{C}]^{2+}\cdot 2\text{e}^-$ electride comes from quasi-atoms of IAEs. To prove that the

ferromagnetic anionic electrons play a key role in ferromagnetic properties of $[\text{Gd}_2\text{C}]^{2+}\cdot 2\text{e}^-$, we performed the DFT calculation for $[\text{Gd}_2\text{C}]^{2+}\cdot 2\Box$ where anionic electrons are removed from $[\text{Gd}_2\text{C}]^{2+}\cdot 2\text{e}^-$ electrone. As shown in Extended Data Table 2, the magnetic moment of $[\text{Gd}_2\text{C}]^{2+}\cdot 2\Box$ in the absence of IAEs is $14.2 \mu_B$ that is much smaller than the value of $15.3 \mu_B$ (DFT) of $[\text{Gd}_2\text{C}]^{2+}\cdot 2\text{e}^-$ electrone, in which the magnetic moments of IAE are $0.52 \mu_B$. Importantly, it should be noted that the $[\text{Gd}_2\text{C}]^{2+}\cdot (1-x)2\text{e}^- \cdot \text{Cl}_x$ with the same Gd lattice showed the antiferromagnetism, which was also confirmed from the theoretical calculations, as shown in Fig. 4.

To verify the comments, we also attempted to construct a “strained” Gd in the mold of $[\text{Gd}_2\text{C}]^{2+}\cdot 2\text{e}^-$ by making a structure of Gd atoms in the position of $[\text{Gd}_2\text{C}]^{2+}\cdot 2\text{e}^-$ without C atoms to compute its magnetic moment and compare it with that of $[\text{Gd}_2\text{C}]^{2+}\cdot 2\text{e}^-$. However, this hypothetical structure is energetically unstable with a much higher formation energy ($+0.105 \text{ eV}$ per Gd atom compared to -0.382 eV per Gd atom of $[\text{Gd}_2\text{C}]^{2+}\cdot 2\text{e}^-$) and did not produce a physically meaningful result. **This result is added in the revised manuscript to provide a clear understanding on the role of IAEs for constructing layer structured ionic crystal and imparting the magnetic properties of $[\text{Gd}_2\text{C}]^{2+}\cdot 2\text{e}^-$ electrone.** To further clarify the role of anionic electron in the ferromagnetic ordering of $[\text{Gd}_2\text{C}]^{2+}\cdot 2\text{e}^-$, we carried out the substitution of Cl atoms in $[\text{Gd}_2\text{C}]^{2+}\cdot 2\text{e}^-$ electrone to achieve the effect of removing anionic electrons experimentally. Our experiment showed that ferromagnetic $[\text{Gd}_2\text{C}]^{2+}\cdot 2\text{e}^-$ electrone to attain antiferromagnetic phase as anionic electrons are absorbed by Cl ions. At the Cl concentration $x=1.0$, or $[\text{Gd}_2\text{C}]^{2+}\cdot (1-x)2\text{e}^- \cdot \text{Cl}_x$, the local magnetic moment of Gd atom is again $7.1 \mu_B$ below the value for Gd metal. Therefore, we can conclude that the quasi-atoms of IAEs play a crucial role in the magnetism of $[\text{Gd}_2\text{C}]^{2+}\cdot 2\text{e}^-$ electrone.

Comparing the distance between Gd-Gd in $[\text{Gd}_2\text{C}]^{2+}\cdot 2\text{e}^-$ and Gd metal, 3.97 \AA of Interlayer distance in $[\text{Gd}_2\text{C}]^{2+}\cdot 2\text{e}^-$ is longer than 3.57 \AA of Gd metal distance. Generally, in magnetic material, as the distance between the magnetic elements increase, the exchange interaction declines and the magnetic properties also decrease. However, even if the distance between the Gd in the interlayer was longer, it was confirmed that the magnetic property of $[\text{Gd}_2\text{C}]^{2+}\cdot 2\text{e}^-$ is stronger than Gd metal. As a result, the magnetic properties of $[\text{Gd}_2\text{C}]^{2+}\cdot 2\text{e}^-$ cannot be attributed to the enhanced magnetic properties with only Gd belonging to the lattice as in general magnetic materials. Experimental results show that IAEs are located in $\sim 4 \text{ \AA}$ interlayer space and Gd-IAE-Gd single magnetic layer is formed. These results show that the improved magnetic properties of $[\text{Gd}_2\text{C}]^{2+}\cdot 2\text{e}^-$ are the result of IAEs mediating the magnetic interaction between the magnetic Gd ions and enhancing the exchange interaction between them.

4. Although evidences like XRD shows the quality of the crystal, we know XRD has detection limit and cannot rule out all impurity phase or defects especially in low concentration (e.g., <0.1%). But throwing the bulk crystals as a whole into magnetometers for measurements may not be careful enough: even <0.1% foreign species or phase can give appreciable magnetic signals given the large chunk of samples. In fact, this is a major source of artifacts that have been routinely measured days after days in worldwide labs. The authors need solid evidence to rule out this scenario, if they insist their claims.

Ans. 4) Yes, we agree with the comments. We have always checked the effect of possible impurities by the chemical analysis using Inductively Coupled Plasma (ICP) spectroscopy. As mentioned by reviewer, in order to avoid the possible contribution from magnetic impurities, the ICP measurements have been conducted as shown in the Table R3. Two Gd raw metal chunks and six [Gd₂C]²⁺·2e⁻ electride flakes were analyzed. The results clearly showed the extremely low concentration of all impurities. The Tb and Er with the highest concentration (around 1ppm) among the analyzed impurities have a ferromagnetic transition temperature 220 K and 19 K, much lower than 350 K of [Gd₂C]²⁺·2e⁻ electride. These results are explained in the Methods section and added in the Supplementary Table 5.

Table R3. Results of inductively coupled plasma spectroscopy measurement for Gd raw material and $[\text{Gd}_2\text{C}]^{2+}\cdot 2\text{e}^-$ electrode.

Ferromagnetic elements

Sample	Fe	Co	Ni	Tb	Dy	Ho	Er	Tm
Gd								
Sample_1	1.411	0.215	0.033	0.800	0.264	0.047	0.009	0.110
Sample_2	2.447	0.089	0.052	0.739	2.477	0.288	1.695	0.079
$[\text{Gd}_2\text{C}]^{2+}\cdot 2\text{e}^-$								
Sample_1	0.501	0.004	0.014	1.077	1.042	0.230	1.670	0.113
Sample_2	None	0.001	0.004	0.741	0.720	0.170	1.269	0.075
Sample_3	0.080	0.007	0.016	1.101	1.122	0.275	2.110	0.119
Sample_4	0.017	0.007	0.012	1.325	1.369	0.326	2.487	0.144
Sample_5	0.691	0.040	0.044	1.035	0.136	0.046	0.016	0.147
Sample_6	0.419	0.024	0.047	0.857	0.127	0.048	0.012	0.121
Average								
	Fe	Co	Ni	Tb	Dy	Ho	Er	Tm
Gd	1.929	0.152	0.043	0.770	1.370	0.167	0.852	0.095
$[\text{Gd}_2\text{C}]^{2+}\cdot 2\text{e}^-$	0.341	0.013	0.022	1.022	0.752	0.182	1.260	0.119

Antiferromagnetic elements

Sample	Cr	Mn	Ce	Nd	Sm
Gd					
Sample_1	0.608	1.329	0.224	0.619	0.742
Sample_2	0.216	0.104	3.031	3.455	0.479
$[\text{Gd}_2\text{C}]^{2+}\cdot 2\text{e}^-$					
Sample_1	0.151	None	0.216	5.031	0.004
Sample_2	0.054	None	0.114	2.803	0.002
Sample_3	0.093	0.002	0.207	5.369	0.003
Sample_4	0.098	0.016	0.233	5.812	0.003
Sample_5	0.355	0.006	0.103	0.881	0.004
Sample_6	0.210	0.063	0.110	0.860	0.004
Average					
	Cr	Mn	Ce	Nd	Sm
Gd	0.412	0.716	1.627	2.037	0.611
$[\text{Gd}_2\text{C}]^{2+}\cdot 2\text{e}^-$	0.160	0.021	0.163	3.459	0.003

Concentration [ppm]
Error range [3%]

Reviewers' Comments:

Reviewer #1:

Remarks to the Author:

The authors have thoroughly revised their manuscript based on the review comments. This paper demonstrated a new family of very unconventional magnetic materials based on electrider compounds, which is quite surprising progress in both magnetism and electrider areas. I believe the marriage of the two otherwise separate areas will lead to a new route toward novel magnetic materials. Furthermore, the studied material is layered and has the potential to become an unusual two-dimensional ferromagnetic material. Therefore, I would like to recommend the publication of the manuscript in Nature Communication.

Reviewer #2:

Remarks to the Author:

The authors' reply appears not convincing. Additional evidence does not give strong support. Some arguments are premature.

1 It appears that the authors do not catch the significance and the in-depth physics of the word "two-dimensional". "Two-dimensional" means the physical dimension along the third direction is significantly shorter than the (electronic, magnetic, or else) correlation length. The authors used the van der Waals structure and the anisotropic electric resistivity (in-plane versus cross-plane) to show it is "two-dimensional". This is incorrect; people never called graphite "two-dimensional". Although the authors admitted the wrong usage by this terminology in rebuttal letter, "two dimensional" is still frequently used throughout the manuscript, including in the title. Because the real two-dimensional material with in-plane magnetic anisotropy cannot support any finite temperature long-range magnetic order, the authors' claim may give a misleading impression they break the Mermin-Wagner theorem and found the "two-dimensional room temperature ferromagnetism in the easy-plane 2D system". This is certainly wrong. Please carefully claim on this.

I strongly believe even if the measurements are all correct, the observation is surely arising from a 3D magnet rather than 2D magnet, just like graphite versus graphene.

2. Using DFT calculations with and without electrons to prove "the magnetism is from the electrider rather than from the Gd ions". This is another wrong approach to prove. Assuming all calculations are correct, the system with added electron concentrations showing magnetism can only mean the doped (or charged) material shows magnetism, which is a common phenomenon. It is incorrect to use this evidence to claim the magnetism is from the added electrons. It is wrong to separate electrons and ionic crystal structure for consideration. Strained structure as a control calculation is improper too.

Here the authors have clearly been aware of that Gd is a magnetic species. Even if the neutral material (without added electrons) is antiferromagnetic as the authors' calculations show, adding electrons to convert the system to be ferromagnetic is a very common approach in many other systems such as MnPS₃. We definitely cannot say that: because the doping converts the MnPS₃ from AFM to FM, the magnetism in doped MnPS₃ is from the added electrons, so the doped MnPS₃ is a 2D electrider. This is incorrect logic.

In summary, I am hesitant to the claim the authors gave. There is a clear shortage of solid evidence.

Response letter to reviewer comments

Reviewer #1 (Remarks to the Author):

The authors have thoroughly revised their manuscript based on the review comments. This paper demonstrated a new family of very unconventional magnetic materials based on electride compounds, which is quite surprising progress in both magnetism and electride areas. I believe the marriage of the two otherwise separate areas will lead to a new route toward novel magnetic materials. Furthermore, the studied material is layered and has the potential to become an unusual two-dimensional ferromagnetic material. Therefore, I would like to recommend the publication of the manuscript in Nature Communication.

→ We greatly appreciate your insightful comments for a promising future of magnetic electrides. Indeed, your expectation for “the potential to become an unusual two-dimensional ferromagnetic material” is being realized in newly discovered van der Waals type $[\text{RECl}]^{2+}\cdot 2\text{e}^-$ electrides. This work is now being prepared to be submitted and the concept “ferromagnetic quasi-atomic anionic electrons to induce the magnetism in electrides” is basically the same as that of the present work. Though the present ferromagnetism of two-dimensional layered structure $[\text{Gd}_2\text{C}]^{2+}\cdot 2\text{e}^-$ electride is not of two-dimensional ferromagnetism, the ideal two-dimensional magnetism can be possible in such van der Waals type electrides, which will soon be reported elsewhere.

Again, we appreciate your positive evaluation and considerations.

Figure R1. Anionic electrons occupying interlayer (left) and intralayer (right) space.

Reviewer #2 (Remarks to the Author):

The authors' reply appears not convincing. Additional evidence does not give strong support. Some arguments are premature.

1. It appears that the authors do not catch the significance and the in-depth physics of the word "two-dimensional". "Two-dimensional" means the physical dimension along the third direction is significantly shorter than the (electronic, magnetic, or else) correlation length. The authors used the van der Waals structure and the anisotropic electric resistivity (in-plane versus cross-plane) to show it is "two-dimensional". This is incorrect; people never called graphite "two-dimensional". Although the authors admitted the wrong usage by this terminology in rebuttal letter, "two dimensional" is still frequently used throughout the manuscript, including in the title.

Because the real two-dimensional material with in-plane magnetic anisotropy cannot support any finite temperature long-range magnetic order, the authors' claim may give a misleading impression they break the Mermin-Wagner theorem and found the "two-dimensional room temperature ferromagnetism in the easy-plane 2D system". This is certainly wrong. Please carefully claim on this.

I strongly believe even if the measurements are all correct, the observation is surely arising from a 3D magnet rather than 2D magnet, just like graphite versus graphene.

→ As we clearly stated in the title "Ferromagnetic quasi-atomic electrons in two-dimensional electride", we do not make any claim on "two-dimensional ferromagnetism" such as the ferromagnetism of van der Waals layered materials, but rather on the role of ferromagnetic quasi-atomic anionic electrons of two-dimensional electride to induce the ferromagnetism.

Since the first discovery of two-dimensional electride by our research groups, $[\text{Ca}_2\text{N}]^+\cdot\text{e}^-$ (Nature, **494**, 336 (2013)), which has anionic electrons occupying two-dimensional interlayer space, many two-dimensional layer structured electrides have been reported experimentally and theoretically, as listed below. Because the anionic electrons are occupying the two-dimensional interlayer space, the two-dimensional electrides show anisotropic physical properties depending on the degree of localization of anionic electrons. This is why the

electride society has adopted the terminology of “two-dimensional electride” in a broad sense.

- (1) Lee, K. et al. Dicalcium nitride as a **two-dimensional electride** with an anionic electron layer. *Nature* **494**, 336–340 (2013).
- (2) Inoshita, T. et al, Exploration for **Two-Dimensional Electrides** via Database Screening and Ab Initio Calculation. *Phys. Rev. X* **4**, 031023 (2014).
- (3) Zhang, X. et al, **Two-Dimensional Transition-Metal Electride** Y_2C . *Chem. Mater.* **26**, 6638 (2014).
- (4) Motoaki Hirayama et al. **Two-dimensional Electrides** as a New Platform of Topological Materials. *Phys. Rev. X* **8**, 031067 (2018).
- (5) Shan Guan, et al. Electronic, Dielectric, and Plasmonic Properties of **Two-Dimensional Electride** Materials X_2N ($X=Ca, Sr$): A First-Principles Study. *Scientific Reports* **5**, Article number 12285 (2015).
- (6) Tomofumi Tada, et al High-Throughput ab Initio Screening for **Two-Dimensional Electride**. *Inorg. Chem.* **53**, 10347–10358 (2014).
- (7) Wnmei. et al. First-Principles Prediction of Thermodynamically Stable **Two-Dimensional Electrides**. *J. Am. Chem. Soc.* **138**, 15336–15344 (2016).
- (8) Jongho Park, et al. Strong Localization of Anionic Electrons at Interlayer for Electrical and Magnetic Anisotropy in **Two-Dimensional** Y_2C Electride. *J. Am. Chem. Soc.* **139**, 615–618 (2017).
- (9) Jongho Park, et al. Tuning the Spin-Alignment of Interstitial Electrons in **Two-Dimensional** Y_2C Electride via Chemical Pressure. *J. Am. Chem. Soc.* **139** 17277–17280 (2017).
- (10) Jianhua Hou et al. **Two-Dimensional** Y_2C Electride: A Promising Anode Material for NaIon Batteries. *J. Phys. Chem. C* **120** (33), 18473–18478 (2016).
- (11) Ye Ji Kim et al. **Two dimensional inorganic electride**-promoted electron transfer efficiency in transfer hydrogenation of alkynes and alkenes. *Chem. Sci.* **6**, 3577–3581 (2015).
- (12) Pierluigi Cudazzo et al. Collective charge excitations of the **two-dimensional electride** Ca_2N . *Phys. Rev. B* **96**, 125131 (2017).

- (13) Yunweu Zhang, et al. Computer-Assisted Inverse Design of Inorganic Electrides. *Phys. Rev. X* **7**, 011017 (2017).
- (14) Biao Wan et al. Identifying quasi-2D and 1D electrides in yttrium and scandium chlorides via geometrical identifications. *NPJ Comp. Mater.* **4**, 77 (2018).
- (15) Bohayra Mortazavi et al. Mechanical, optoelectronic and transport properties of single-layer Ca_2N and Sr_2N electrides. *J. Alloys Compd.* **739**, 643–652 (2018).
- (16) Daniel L. Druffel et al. Electrons on the surface of 2D materials from layered electrides to 2D electrenes. *J. Mater. Chem. C*, **5**, 11196–11213 (2017).
- (17) Junjie Wang et al. Ternary inorganic electrides with mixed bonding. *Phys. Rev. B* **99**, 064104 (2019).
- (18) Stephen G. Dale et al. Theoretical Descriptors of Electrides. *J. Phys. Chem. A* **122**, 9371–9391 (2018).
- (19) Dandan Wang et al. First-principles study on OH-functionalized 2D electrides: Ca_2NOH and $\text{Y}_2\text{C}(\text{OH})_2$, promising two-dimensional monolayers for metal-ion batteries. *Appl. Surf. Sci.* **478**, 459–464 (2019).
- (20) Youngtek Oh et al. Electric field effect on the electronic structure of 2D Y_2C electride. *2D Mater.* **5**, 035005 (2018).

As shown in Supplementary Figure 1 of the revised manuscript, the distance (3.63 Å in $[\text{Gd}_2\text{C}]^{2+}\cdot 2\text{e}^-$) between quasi-atomic anionic electrons in the in-plane direction is different from the distance (2.68 Å in $[\text{Gd}_2\text{C}]^{2+}\cdot 2\text{e}^-$) between quasi-atomic anionic electrons and cations (Gd ions) in the out-of-plane direction, indicating that anisotropic physical properties are attributed to the existence of anionic electrons. Furthermore, the layer structure of “two-dimensional electride” is distinctively different from that of the van der Waals two-dimensional materials because anionic electrons occupy the interlayer spacing as constituent ions of exotic ionic electride, while there are no ions in the interlayer spacing of the van der Waals layered materials. This makes two-dimensional electrides with inherent anionic electrons different from the van der Waals two-dimensional materials.

Figure R1. Anionic electrons occupying interlayer (left) and intralayer (right) space.

We agree with your assertion on the ferromagnetism of the present system, which is definitely not of two-dimensional ferromagnetism as reported in the van der Waals layered ferromagnetic materials. In fact, to demonstrate this point, we considered additional exchange interaction between Gd atoms across C atoms of intralayer slab in out-of-plane direction for more accurate analysis as shown in Supplementary Figure 9 of the revised manuscript. And the magnetic transition temperatures are recalculated and showed in revised Figure 4. Although the exchange interaction of Gd-C-Gd (J_3 , exchange interaction of out-of-plane Gd-Gd atoms across C atoms in the intralayer slab) is smaller than that of Gd- e^- -Gd in the out-of-plane direction (J_2 , exchange interaction of out-of-plane Gd-Gd atoms across quasi-atomic anionic electrons in the interlayer space), it is not negligible and it clearly show that the ferromagnetism of $[\text{Gd}_2\text{C}]^{2+}\cdot 2e^-$ electride is not two-dimensional but three-dimensional as we stated in our previous revision and the present revision. This clearly implies a three-dimensional ferromagnetism in two-dimensional layer structured $[\text{Gd}_2\text{C}]^{2+}\cdot 2e^-$ electride. However, the anisotropic magnetic property of two-dimensional $[\text{Gd}_2\text{C}]^{2+}\cdot 2e^-$ electride is an undeniable fact. Therefore, respecting your concerns on two-dimensional ferromagnetism, we revised our manuscript more carefully to avoid any possible misunderstanding and to make it more clear that the present ferromagnetism arises three-dimensionally rather than two-dimensionally.

Additionally, we'd like to introduce shortly about another two-dimensional electride (not

reported yet) that our group discovered, which has different occupations of quasi-atomic anionic electrons. In previously reported two-dimensional electrides such as $[\text{Ca}_2\text{N}]^+\cdot\text{e}^-$, and $[\text{Y}_2\text{C}]^{2+}\cdot 2\text{e}^-$ and $[\text{Gd}_2\text{C}]^{2+}\cdot 2\text{e}^-$, the anionic electrons are occupying two-dimensional interlayer spacing. Whereas, in the newly discovered two-dimensional electride, the anionic electrons are occupying two-dimensional intralayer spacing as schematically illustrated in Figure R1. First of all, the newly discovered two-dimensional electride is a van der Waals type layered material. Moreover, this van der Waals type electride showed the two-dimensional magnetism as previously reported van der Waals 2D magnets did. Since our group also have studied the van der Waals layered materials (Nature Physics, **11**, 482 (2015); Science, **349**, 625 (2015); Nature Physics, **13**, 931 (2017)), we have recognized the “two-dimensional van der Waals ferromagnet”. Thus, we knew that the ferromagnetism of $[\text{Gd}_2\text{C}]^{2+}\cdot 2\text{e}^-$ electride was not of the two-dimensional ferromagnetism, either.

We appreciate your constructive comments to make this report clearer.

2. Using DFT calculations with and without electrons to prove “the magnetism is from the electrider rather than from the Gd ions”. This is another wrong approach to prove. Assuming all calculations are correct, the system with added electron concentrations showing magnetism can only mean the doped (or charged) material shows magnetism, which is a common phenomenon. It is incorrect to use this evidence to claim the magnetism is from the added electrons. It is wrong to separate electrons and ionic crystal structure for consideration. Strained structure as a control calculation is improper too.

Here the authors have clearly been aware of that Gd is a magnetic species. Even if the neutral material (without added electrons) is antiferromagnetic as the authors’ calculations show, adding electrons to convert the system to be ferromagnetic is a very common approach in many other systems such as MnPS3. We definitely cannot say that: because the doping converts the MnPS3 from AFM to FM, the magnetism in doped MnPS3 is from the added electrons, so the doped MnPS3 is a 2D electrider. This is incorrect logic.

→ 1) Again, we’d like to emphasize that the nature of electrideres is totally different from that of the typical materials discussed in the scheme of orbital electrons.

The fundamentals: i) what the electrider is, and ii) what the difference between interstitial anionic electrons (IAEs) in electrider and electrons in non-electrider systems is, are the most important to understand what we report on the $[\text{Gd}_2\text{C}]^{2+}\cdot 2\text{e}^-$ electrider.

Electrideres are stoichiometric ionic crystals in which electrons are trapped in cavities and serve as the counteranions to an equal number of positive charges in a regular crystalline array. These anionic electrons occupy the real space (interstitial sites with the size of ~ 0.4 nm) of lattice framework, not the orbitals of constituent atoms. Thus, the electrons in electrideres are referred as the interstitial anionic electrons (IAEs) and behave as quasi-atoms like building components (constituent atoms of materials), which are different from the orbital electrons of constituent atoms in typical materials.

It should be noted that DFT calculations of electrideres have been performed without intentional or artificial addition of electrons into the systems. In general, DFT calculations performed with experimentally determined chemical composition and crystal structure naturally results in anionic interstitial electrons occupying the interlayer space that can be confirmed by electron

localization function (ELF) or charge density plots. Obviously, this is a common way to theoretically calculate the electronic structures of electrides and their properties. Furthermore, in most of theoretical studies on elemental electrides under high pressure, the formation of anionic electrons that are coordinated at structural sites is naturally deduced without intentional addition of electrons into the transformed structure.

→ 2) We claim that the ferromagnetism of $[\text{Gd}_2\text{C}]^{2+}\cdot 2\text{e}^-$ electride originates from both ferromagnetic quasi-atomic anionic electrons and Gd ions, not just from Gd ions. Importantly, we explained the crucial role of quasi-atomic anionic electrons in the ferromagnetic $[\text{Gd}_2\text{C}]^{2+}\cdot 2\text{e}^-$ electride. Thus, we provided the theoretical and experimental evidence for anionic electron-removed systems and chlorine-substituted systems, in which both systems have the same arrangement of Gd ions. In experiments, when the ferromagnetic quasi-atomic anionic electrons are substituted with the paramagnetic chlorines, the antiferromagnetism occurs. This experimental result strongly indicates that the arrangement of Gd ions in $[\text{Gd}_2\text{C}]^{2+}\cdot 2\text{e}^-$ electride can not result in the ferromagnetism. However, the coexistence of quasi-atomic anionic electrons and chlorine atoms showed the ferromagnetism, which has a lower ferromagnetic transition temperature than that of $[\text{Gd}_2\text{C}]^{2+}\cdot 2\text{e}^-$ electride. Furthermore, the theoretical calculations for the quasi-atomic anionic electrons-removed system, which has the same arrangement of Gd ions as chlorine-substituted system, also showed the antiferromagnetism. Thus, we explained that the exchange interaction between ferromagnetic quasi-atomic anionic electrons and Gd ions was crucial for the emergence of ferromagnetism in $[\text{Gd}_2\text{C}]^{2+}\cdot 2\text{e}^-$ electride. Thus, it is clear that the ferromagnetism originates from the inherent interstitial anionic electrons, which is the characteristic feature of an electride, not from the added electrons. This experimental demonstration of ferromagnetic electride based on the crucial role of ferromagnetic quasi-atomic anionic electrons is the first report in the electride research field and will be an important report to further develop magnetic electrides, in particular low-dimensional ferromagnetic electrides as we introduced the van der Waals type magnetic electride in the above response.

→ 3) We understand the example about the transition from AFM and to FM with additional electrons. However, as mentioned above, the electrides have inherent interstitial anionic

electrons as constituent ions, not intentionally added electrons. If we added the electrons in the typical materials such as MnPS_3 via electrostatical, physical or chemical methods, the added electrons will occupy the orbitals of constituent atoms. This is the most different aspect from the interstitial anionic electrons, which occupy the real space of lattice in electrides. Thus, your comment that “the electron-doped MnPS_3 is not electride” is absolutely right. However, if the doped electrons in MnPS_3 , that is added electrons, can occupy the structural real space in the lattice and can work as anions, the MnPS_3 with anionic electrons occupying real space can be an electride. This will not be realized. In other words, our logic started from the electride with inherent anionic electrons occupying structural sites, which have been referred as quasi-atoms in electrides.

In summary, I am hesitant to the claim the authors gave. There is a clear shortage of solid evidence.

→ We understand that our logical claims could appear unusual from the viewpoint of typical materials. Moreover, regarding your main argument on the “two-dimensional ferromagnetism”, although we did not claim the two-dimensional ferromagnetism like van der Waals ferromagnetic layered materials, we admit that we could have been more clear in describing two-dimensional electrides and the ferromagnetism of $[\text{Gd}_2\text{C}]^{2+}\cdot 2\text{e}^-$ electrides. Therefore, we revised the manuscript more carefully to avoid the misunderstanding or misreading of our main claims. We hope that the above explanations on the unusual properties of this unique electride are satisfactory and make our manuscript acceptable for publication in Nature Communications.

Again, we appreciate your constructive comments to make this report on the first demonstration of ferromagnetic electride better and clearer.

Reviewers' Comments:

Reviewer #2:

Remarks to the Author:

I am not convinced by the reply to the previous comments.

Reviewer #3:

Remarks to the Author:

This work synthesised and examined a new 2D electride $GdC_{2+} : 2e^-$ and found evidence of 3D ferromagnetic properties, the first experimental electride work to do so. I believe this work has significant potential impact and qualifies for publication in Nature Communications. I do however have a number of concerns, listed below, that I believe should be addressed before publication.

1) The article briefly discusses previous instances of theoretically predicted magnetic behaviour in electrides, but misses a few examples.

I think it important to include:

``Ferromagnetic instability of interlayer floating electrons in the quasi-two-dimensional electride Y_2C' which predicts itinerant ferromagnetism in Y_2C , a material isostructural to Gd_2C .

``Pressure-Induced Isostructural Antiferromagnetic–Ferromagnetic Transition in an Organic Electride' which predicts ferromagnetic behaviour under pressure in an organic electride, adding to the sentence pointing out the organic electrides possess antiferromagnetic behaviour (line 96).

2) The Bader charge analysis in this work is unusual using the magnetization density rather than total electron density. This is not standard practise and does not appear ref 30 which is cited to justify this approach. To the best of my knowledge this approach is coined quantum chemical topology, primarily in work by Paul Popelier. The first instance of Bader charge analysis being in this work used is on line 217 and I believe the specific approach to the charge analysis, and background for this approach, should be highlighted there.

Further, Supp Fig 6a shows the Bader basin volumes used in this work. The Bader basin of the interstitial electron clearly overlaps with the Gd atom. I suspect this is a symptom of a 2D slice being cut out of a 3D density plot, however this a) needs to be clarified (and I would be extremely concerned if the Bader basin does indeed overlap with the Gd atom) and b) further highlights the specific approach taken to Bader charge analysis in this work.

3) In general, the computational methods used in this work are not well specified.

- The PBE functional should be accompanied by a citation in addition to the Kresse implementation.

- The specific programs (and versions) used for calculation and subsequent analysis should be highlighted and accompanied by citations

- The ELF method should be accompanied by a citation

- the PAW method should be accompanied by a citation

- Is quadruple replications of the unit cell sufficient for an G-type anti-ferromagnetic description? Assuming 1 magnetic moment per unit-cell, 8 replication would be necessary.
- Inclusion of the specific geometries used to conduct the calculations in this work in the SI would be highly desirable.

4) In the paragraph between lines 239 and 257, it is unclear whether the results being discussed are experimental or theoretical. Further, in Supp Fig 7 it is unclear what the x's are designating, this should be specified in the figure caption (granted this is done so in previous figures, each figure should be able to stand alone). Also, the bonds between Gd and C look like the x being used to highlight interstitial electron, causing some confusion. Could this distinction be made more clear?

5) Lines 268-288 discuss exchange interaction coupling constants obtained from DFT calculations. This is not a sufficient explanation for how these coupling constants were obtained and more specific details are necessary.

6) ``three dimensional' appears a few times in the text without the accompanying (3D) abbreviation that 2D and 1D receive. I found this odd.

I look forward to seeing this work reach print and future electricle discoveries from the authors.

Best,

Stephen Dale

Reviewer #4:

Remarks to the Author:

The authors have done a good job of answering the referee's concerns. Indeed referee two even misstates the Mermin-Wagner theorem, which strictly speaking only applies to a rotationally invariant system with short-range magnetic interactions. 2D easy plane ferromagnets exists and do not violate the theorem.

As such I support publication of the manuscript.

Response letter to reviewer comments

Reviewers' comments:

Reviewer #3 (Remarks to the Author):

This work synthesized and examined a new 2D electride $\text{Gd}_2\text{C}^{2+}\cdot 2\text{e}^-$ and found evidence of 3D ferromagnetic properties, the first experimental electride work to do so. I believe this work has significant potential impact and qualifies for publication in *Nature Communications*. I do however have a number of concerns, listed below, that I believe should be addressed before publication.

Ans.) We greatly appreciate your positive evaluations. We do concur with your insight that the ferromagnetic coupling of the local spins mediated by the anionic electrons acting as quasi-atoms will be a new interesting phenomenon in 2D materials beyond electrides. Following your valuable comments, we have verified your suggestions and revised the manuscript for the ferromagnetism of $[\text{Gd}_2\text{C}]^{2+}\cdot 2\text{e}^-$ electride.

1) The article briefly discusses previous instances of theoretically predicted magnetic behavior in electrides, but misses a few examples. I think it important to include: “Ferromagnetic instability of interlayer floating electrons in the quasi-two-dimensional electride Y_2C ” which predicts itinerant ferromagnetism in Y_2C , a material isostructural to Gd_2C . “Pressure-Induced Isostructural Antiferromagnetic–Ferromagnetic Transition in an Organic Electride” which predicts ferromagnetic behavior under pressure in an organic electride, adding to the sentence pointing out the organic electrides possess antiferromagnetic behavior (line 96).

Ans.) We added the references (Refs. 22 and 23) in the relevant parts of the Introduction. Also, we modified the related sentences as followings; The ferromagnetic instability is also expected in the 2D layer structured inorganic $[\text{Y}_2\text{C}]^{2+}\cdot 2\text{e}^-$ electride²². Furthermore, it has been demonstrated that the localized IAEs of several organic electrides interact with each other through cavities in a one-dimensional (1D) channel, rendering a weak magnetic ordering such as antiferromagnetism². It is further predicted that the antiferromagnetic-ferromagnetic transition in the simplest organic $\text{Cs}^+(15\text{-crown-5})_2\cdot 2\text{e}^-$ electride can be realized under an easily

accessible experimental condition (0.5–1 GPa) which enables a strong spin coupling of localized IAEs²³. However, to date, a ferromagnetic electride has not been discovered in experiments.

Considering the antiferromagnetism and possible ferromagnetism of organic electrides originating from IAEs in 1D channels together with the anisotropy of electronic states^{2,23,24}, it is possible to realize a ferromagnetic electride based on strongly localized IAEs occupying a regular array in interlayer space of layer structured 2D electride.

2) The Bader charge analysis in this work is unusual using the magnetization density rather than total electron density. This is not standard practice and does not appear ref 30 which is cited to justify this approach. To the best of my knowledge this approach is coined quantum chemical topology, primarily in work by Paul Popelier. The first instance of Bader charge analysis being in this work used is on line 217 and I believe the specific approach to the charge analysis, and background for this approach, should be highlighted there.

Ans.) Thank you very much for the comments. As you suggested, we revised the manuscript to include the specific approach and background to our magnetic moment analysis in the Method section. We also added Popelier's work as Ref. 34. We also highlighted the advantage of the Bader decomposition method over the conventional projection method. The revised portion of the Method section now reads:

“The Bader basin for each site is computed as the volume containing a single magnetization density maximum and is separated from other volumes by a zero-flux surface of the gradients of the magnitude of the magnetization density. Once a Bader basin is determined, the atomic magnetic moment is computed by integrating the signed value of magnetization density within the basin. Our calculation shows that this new approach of applying Bader analysis to magnetization density provides more accurate and robust measure of magnetic moment for a system with delocalized IAEs such as $[\text{Gd}_2\text{C}]^{2+}\cdot 2\text{e}^-$ electride than the conventional projection method that strongly depends on the size of artificially set projection spheres and inevitably suffers the problem of undercounting and double counting (see the comparison in Supplementary Fig. 6, Supplementary Tables 3 and 4).”

Regarding a more detailed approach, one can consider the following two methods:

(Method 1) Partition the volume using the *magnitude of magnetization density* and use the *signed value of magnetization density* to compute the local magnetic moment.

(Method 2) Partition the volume using the *charge density* and use the *signed value of magnetization density* to compute the local magnetic moment.

In the present work, we used **Method 1**. In many conventional non-electride systems, **Method 2** can give similarly reasonable local magnetic moments. However, in a system with delocalized IAEs such as the present $[\text{Gd}_2\text{C}]^{2+}\cdot 2\text{e}^-$ electride, **Method 1** gives a much more accurate result consistent with the experiment. Fig. R1 shows why **Method 2** has difficulty in handling the case of electrides. As shown in Fig. R1a, the charge density distribution of $[\text{Gd}_2\text{C}]^{2+}\cdot 2\text{e}^-$ electride is overwhelmingly dominated by Gd atoms (Gd atoms have 18 valence electrons compared to 4 for C atoms and 2 for IAEs), and it is quite difficult to see the presence of IAEs. On the other hand, the ELF (Fig. R1b) and the magnetization density (Fig. R1c) clearly show the presence of the interstitial electrons. This is due to the fact that charge density is the sum of spin-up and spin-down charge densities while magnetization density is the difference of spin-up and spin-down charge densities. If one decomposes the volume based on charge density, the features in the difference of spin-up and spin-down charge densities will be washed out by a large valence charge density of Gd atoms. **Method 2** adopted in the present work gives results consistent with the experiment.

Further, Supp. Fig. 6b shows the Bader basin volumes used in this work. The Bader basin of the interstitial electron clearly overlaps with the Gd atom. I suspect this is a symptom of a 2D slice being cut out of a 3D density plot, however this a) needs to be clarified (and I would be extremely concerned if the Bader basin does indeed overlap with the Gd atom) and b) further highlights the specific approach taken to Bader charge analysis in this work.

Ans.) As the Reviewer pointed out, it would be problematic if the Bader basin of the IAEs overlaps with the Gd atom. We clarify in the most certain terms that it does not. We acknowledge that the Supplementary Fig. 6b does give such an impression because the figure contained a full 3D atomic structure of $[\text{Gd}_2\text{C}]^{2+}\cdot 2e^-$ with the slice plane of Bader basin: the Gd atoms that appear to overlap with the Bader basins of the interstitial electron are actually off the slice plane, and they are not inside of any of the Bader basins of interstitial electrons. To avoid such misreading, the atomic configuration of $[\text{Gd}_2\text{C}]^{2+}\cdot 2e^-$ near (110) plane with different angles were displayed in Supplementary Fig. 6a.

3) In general, the computational methods used in this work are not well specified.

- The PBE functional should be accompanied by a citation in addition the Kresse implementation.

Ans.) Added as Ref. 36 and 37.

- The specific programs (and versions) used for calculation and subsequent analysis should be highlighted and accompanied by citations

Ans.) Revised the Method section to explicitly cite the program. Revised Ref. 38 to cite the correct version of the code used.

- The ELF method should be accompanied by a citation

Ans.) Added as Ref. 32.

- the PAW method should be accompanied by a citation

Ans.) Added as Ref. 37.

- Is quadruple replications of the unit cell sufficient for an G-type anti-ferromagnetic description? Assuming 1 magnetic moment per unit-cell, 8 replications would be necessary.

Ans.) We added Supplementary Fig. 11 to show the specific geometries of the magnetic configurations used for the computation of exchange parameters. The Reviewer is absolutely correct that eight independent configurations are required to represent full A-, C-, and G-type antiferromagnetic descriptions. Since there are two Gd atoms in one formula unit of $[\text{Gd}_2\text{C}]^{2+}\cdot 2\text{e}^-$, we only need to extend the primitive unit cell four times.

- Inclusion of the specific geometries used to conduct the calculations in this work in the SI would be highly desirable.

Ans.) We added Supplementary Fig. 11 to show the specific geometries of the magnetic configurations used for the computation of exchange parameters.

4) In the paragraph between lines 239 and 257, it is unclear whether the results being discussed are experimental or theoretical.

Ans.) Thank you for the comment that makes a better readability. As you pointed out, because

the paragraph deals with two systems of Cl-substituted $[\text{Gd}_2\text{C}]^{2+} \cdot (1-x)2\text{e}^- \cdot \text{Cl}_x$ and IAE-removed $[\text{Gd}_2\text{C}]^{2+} \cdot y\Box \cdot (2-y)\text{e}^-$, it was difficult to figure out the theoretical and experimental results. Thus, to make this point clearer, we revised the first sentence of the paragraph as below:

The ferromagnetic quasi-atomic nature of IAEs in $[\text{Gd}_2\text{C}]^{2+} \cdot 2\text{e}^-$ electride was further verified by comparing the experimental characterizations of Cl-substituted $[\text{Gd}_2\text{C}]^{2+} \cdot (1-x)2\text{e}^- \cdot \text{Cl}_x$ system and theoretical calculations of IAE-removed $[\text{Gd}_2\text{C}]^{2+} \cdot y\Box \cdot (2-y)\text{e}^-$ system (\Box represents the vacancy of IAE) that cannot be synthesized in experiments.

Further, in Supp Fig 7 it is unclear what the x's are designating, this should be specified in the figure caption (granted this is done so in previous figures, each figure should be able to stand alone). Also, the bonds between Gd and C look like the x being used to highlight interstitial electron, causing some confusion. Could this distinction be made more clear?

We modified Fig. 7 using the different designation for IAEs instead of simple "X" to make this point clearer, and added the relevant description in the figure caption, as you suggested. In addition, the designation for IAEs in Fig. 3 was also modified to ensure consistency within the whole manuscript.

5) Lines 268-288 discuss exchange interaction coupling constants obtained from DFT calculations. This is not a sufficient explanation for how these coupling constants were obtained and more specific details are necessary.

Ans.) We added Supplementary Fig. 11 to show the specific geometries of the magnetic configurations used for the computation of exchange parameters. We also added the detail of exchange parameter calculations in the Method section and the caption of Supplementary Fig. 11.

6) “three dimensional” appears a few times in the text without the accompanying (3D) abbreviation that 2D and 1D receive. I found this odd.

Ans.) We used the abbreviation “3D” in the manuscript, as you suggested.

I look forward to seeing this work reach print and future electrone discoveries from the authors.

We greatly appreciate your strong support for the publication of our manuscript in *Nature Communications*.

Reviewer #4 (Remarks to the Author):

The authors have done a good job of answering the referee's concerns. Indeed, referee two even misstates the Mermin-Wagner theorem, which strictly speaking only applies to a rotationally invariant system with short-range magnetic interactions. 2D easy plane ferromagnets exist and do not violate the theorem.

As such I support publication of the manuscript.

Ans.) We greatly appreciate your insightful comments and strong support for the publication of our manuscript in *Nature Communications*. Thanks to your support, we can resolve the issue raised by the referee #2.

Reviewers' Comments:

Reviewer #3:

Remarks to the Author:

The authors have sufficiently addressed my comments and I support the publication of this manuscript.